# Online Continuous Hyperparameter Optimization for Generalized Linear Contextual Bandits

**Yue Kang**                                                    *yuekang@ucdavis.edu*
*University of California, Davis*

**Cho-Jui Hsieh**                                               *chohsieh@cs.ucla.edu*
*Google and University of California, Los Angeles*

**Thomas C. M. Lee**                                            *tcmlee@ucdavis.edu*
*University of California, Davis*

**Reviewed on OpenReview:** *https://openreview.net/forum?id=lQE2AcbYge*

## Abstract

In stochastic contextual bandits, an agent sequentially makes actions from a time-dependent action set based on past experience to minimize the cumulative regret. Like many other machine learning algorithms, the performance of bandits heavily depends on the values of hyperparameters, and theoretically derived parameter values may lead to unsatisfactory results in practice. Moreover, it is infeasible to use offline tuning methods like cross-validation to choose hyperparameters under the bandit environment, as the decisions should be made in real-time. To address this challenge, we propose the first online continuous hyperparameter tuning framework for contextual bandits to learn the optimal parameter configuration in practice within a search space on the fly. Specifically, we use a double-layer bandit framework named CDT (Continuous Dynamic Tuning) and formulate the hyperparameter optimization as a non-stationary continuum-armed bandit, where each arm represents a combination of hyperparameters, and the corresponding reward is the algorithmic result. For the top layer, we propose the Zooming TS algorithm that utilizes Thompson Sampling (TS) for exploration and a restart technique to get around the *switching* environment. The proposed CDT framework can be easily utilized to tune contextual bandit algorithms without any pre-specified candidate set for multiple hyperparameters. We further show that it could achieve a sublinear regret in theory and performs consistently better than all existing methods on both synthetic and real datasets.

## 1 Introduction

The contextual bandit is a powerful framework for modeling sequential learning problems under uncertainty, with substantial applications in recommendation systems (Li et al., 2010), clinical trials (Woodroofe, 1979), personalized medicine (Bastani & Bayati, 2020), etc. At each round $t$, the agent sequentially interacts with the environment by pulling an arm from a feasible arm set $\mathcal{A}_t$ of $K$ arms ($K$ might be infinite), where every arm could be represented by a $d$-dimensional feature vector, and only the reward of the selected arm is revealed. Here $\mathcal{A}_t$ is drawn IID from an unknown distribution. In order to maximize the cumulative reward, the agent would update its strategy on the fly to balance the exploration-exploitation tradeoff.

Generalized linear bandit (GLB) was first proposed in Filippi et al. (2010) and has been extensively studied under various settings over the recent years (Jun et al., 2017; Kang et al., 2022), where the stochastic payoff of an arm follows a generalized linear model (GLM) of its associated feature vector and some fixed, but initially unknown parameter $\theta^*$. Note that GLB extends the linear bandit (Abbasi-Yadkori et al., 2011) in representation power and has greater applicability in the real-world applications, e.g. logistic bandit

algorithms Zhang et al. (2016) can achieve improvement over linear bandit when the rewards are binary. Upper Confidence Bound (UCB) (Auer et al., 2002a; Filippi et al., 2010; Li et al., 2010) and Thompson Sampling (TS) (Agrawal & Goyal, 2012; 2013) are the two most popular ideas to solve the GLB problem. Both of these methods could achieve the optimal regret bound of order $\tilde{O}(\sqrt{T})^1$ under some mild conditions, where $T$ stands for the total number of rounds (Agrawal & Goyal, 2013).

However, the empirical performance of these bandit algorithms significantly depends on the configuration of hyperparameters, and simply using theoretical optimal values often yields unsatisfactory practical results, not to mention some of them are unspecified and need to be learned in reality. For example, in both LinUCB (Li et al., 2010) and LinTS (Abeille & Lazaric, 2017; Agrawal & Goyal, 2013) algorithms, there are hyperparameters called exploration rates that govern the tradeoff and hence the learning process. But it has been empirically verified that the best exploration rate to use is always instance-dependent and may vary at different iterations Bouneffouf & Claeys (2020); Ding et al. (2022b). Note it is inherently impossible to use any state-of-the-art offline hyperparameter tuning methods such as cross validation (Stone, 1974) or Bayesian optimization (Frazier, 2018) since decisions in bandits should be made in real time. To choose the best hyperparameters, some previous works use grid search in their experiments (Ding et al., 2021; Jun et al., 2019), but obviously, this approach is infeasible when it comes to reality, and how to manually discretize the hyperparameter space is also unclear. Conclusively, this limitation has already become a bottleneck for bandit algorithms in real-world applications, but unfortunately, it has rarely been studied in the previous literature.

The problem of hyperparameter optimization for contextual bandits was first studied in Bouneffouf & Claeys (2020), where the authors proposed two methods named OPLINUCB and DOPLINUCB to learn the practically optimal exploration rate of LinUCB in a finite candidate set by viewing each candidate as an arm and then using multi-armed bandit to pull the best one. However, 1) the authors did not provide any theoretical support, and 2) we believe the best exploration parameter in practice would vary during iterations – more exploration may be preferred at the beginning due to the lack of observations, while more exploitation would be favorable in the long run when the model estimate becomes more accurate. Furthermore, 3) they only consider tuning one single hyperparameter. To tackle these issues, Ding et al. (2022b) proposed TL and Syndicated framework by using a non-stationary multi-armed bandit for the hyperparameter set. However, their approach still requires a pre-defined set of hyperparameter candidates. In practice, choosing the candidates requires domain knowledge and plays a crucial role in the performance. Also, using a piecewise-stationary setting instead of a complete adversarial bandit (e.g. EXP3) for hyperparameter tuning is more efficient since we expect a fixed hyperparameter setting would yield indistinguishable results in a period of time. Conclusively, it would be more efficient to use a continuous space for bandit hyperparameter tuning.

We propose an efficient bandit-over-bandit (BOB) framework (Cheung et al., 2019) named Continuous Dynamic Tuning (CDT) framework for bandit hyperparameter tuning in the continuous hyperparameter space, without requiring a pre-defined set of hyperparameter candidate configurations. For the top layer bandit we formulate the online hyperparameter tuning as a non-stationary Lipschitz continuum-arm bandit problem with noise where each arm represents a hyperparameter configuration and the corresponding reward is the performance of the GLB, and the expected reward is a time-dependent Lipschitz function of the arm with some biased noise. Here the bias depends on the previous observations since the history could also affect the update of bandit algorithms. It is also reasonable to assume the Lipschitz functions are piecewise stationary since we believe the expected reward would be stationary with the same hyperparameter configuration over a period of time (i.e. *switching* environment). Specifically, for the top layer of our CDT framework, we propose the Zooming TS algorithm with Restarts, and the key idea is to adaptively refine the hyperparameter space and zoom into the regions with more promising reward (Kleinberg et al., 2019) by using the TS methodology (Chapelle & Li, 2011). Moreover, we demonstrate that a simple restart trick could handle the piecewise changes of the bandit environments in both theory and practice. To sum up, we summarize our contributions as follows:

1) We propose an online continuous hyperparameter optimization framework for contextual bandits called CDT that handles all aforementioned issues of previous methods with theoretical guarantees. To the best of our knowledge, CDT is the first hyperparameter tuning method (even model selection method) with

---

$^1\tilde{O}(\cdot)$ ignores the poly-logarithmic factors.

continuous candidates in the bandit community. 2) For the top layer of CDT, we propose the Zooming TS algorithm with Restarts for Lipschitz bandits under the *switching* environment. To the best of our knowledge, our work is the first one to consider the Lipschitz bandits under the *switching* environment, and the first one to utilize TS methodology in Lipschitz bandits. 3) Experiments on both synthetic and real datasets with various GLBs validate the efficiency of our method.

**Notations:** For a vector $x \in \mathbb{R}^d$, we use $\|x\|$ to denote its $l_2$ norm and $\|x\|_A := \sqrt{x^\top A x}$ for any positive definite matrix $A \in \mathbb{R}^{d \times d}$. We also denote $[T] = \{1, \ldots, T\}$ for $T \in \mathbb{N}^+$.

## 2 Related Work

There has been extensive literature on contextual bandit algorithms, and most of them are based on the UCB or TS techniques. For example, several UCB-type algorithms have been proposed for GLB, such as GLM-UCB (Filippi et al., 2010) and UCB-GLM (Li et al., 2017) that achieve the optimal $\tilde{O}(\sqrt{T})$ regret bound. Another rich line of work on GLBs follows the TS idea, including Laplace-TS (Chapelle & Li, 2011), SGD-TS (Ding et al., 2021), etc. In this paper, we focus on the hyperparameter tuning of contextual bandits, which is a practical but under-explored problem. For related work, Sharaf & Daumé III (2019) first studied how to learn the exploration parameters in contextual bandits via a meta-learning method. However, this algorithm fails to adjust the learning process based on previous observations and hence can be unstable in practice. Bouneffouf & Claeys (2020) then proposed OPLINUCB and DOPLINUCB to choose the exploration rate of LinUCB from a candidate set, and moreover Ding et al. (2022b) formulates the hyperparameter tuning problem as a non-stochastic multi-armed bandit and utilizes the classic EXP3 algorithm. However, as we mentioned in Section 1, both works have several limitations that could be decently fixed. Note that hyperparameter tuning could be regarded as a branch of model selection in bandit algorithms. To name a few for this general problem, Agarwal et al. (2017) proposed a master algorithm that could combine multiple bandit algorithms, while Foster et al. (2019) initiated the study of model selection tradeoff in contextual bandits and proposed the first model selection algorithm for contextual linear bandits. Pacchiano et al. (2020) further considered the confidence tuning in OFUL and model selection in reinforcement learning. However, these general model selection methods may fail for the bandit hyperparameter tuning task. To clarify this point, we take the state-of-the-art corralling idea Agarwal et al. (2017) as an example: in theory, it has regret bound or order $O(\sqrt{MT} + MR_{\max})$ where $M$ is the number of base models (number of hyperparameter combinations in our setting) and $R_{\max}$ is the regret of the worst candidate model in the tuning set. Therefore, on the one hand, $M$ is infinitely large in our problem setting with a continuous candidate set, which means the regret bound would also be infinitely large. On the other hand, in order to achieve sub-linear regret in hyperparameter tuning, the corralling idea requires that all hyperparameter candidates yield sub-linear regret in theory, which is a very unrealistic assumption. On the contrary, our work only assumes the existence of a hyperparameter candidate in the tuning set which yields good theoretical regret in theory. In experiments, it is also costly to use since it requires updating all base models at each round, and we have infinitely many base models under our setting. Ding et al. (2022b) includes the corralling idea in their experiments, and we can observe that it achieves almost linear regret in each setting since it has no sub-linear regret guarantee for the bandit hyperparameter tuning problem. In conclusion, the only existing methods that focus on hyperparameter tuning of bandits are OP and TL (Syndicated), and we use both of them in our paper as baselines. And we propose the first continuous hyperparameter tuning framework for contextual bandits, which doesn't require a pre-defined set of candidates. Note it is doable to finely discretize the continuous space and then implement an algorithm with discrete candidate sets (e.g. Syndicated) in methodology, but we highlight the inefficiency of this idea on both the empirical and theoretical side in Appendix A.4.

We also briefly review the literature on Lipschitz bandits that follows two key ideas. One is uniformly discretizing the action space into a mesh (Kleinberg, 2004; Magureanu et al., 2014) so that any learning process like UCB could be directly utilized. Another more popular idea is adaptive discretization on the action space by placing more probes in more encouraging regions (Bubeck et al., 2008; Kleinberg et al., 2019; Lu et al., 2019; Valko et al., 2013), and UCB could be used for exploration. Furthermore, the Lipschitz bandit under adversarial corruption was recently studied in Kang et al. (2023). In addition, (Podimata & Slivkins, 2021) proposed the first fully adversarial Lipschitz bandit in an adaptive refinement manner and derived instance-dependent regret bounds, but their algorithm relies on some unspecified hyperparameters and is

computationally infeasible. Since the expected reward function for hyperparameters would not drastically change every time, it is also inefficient to use a fully adversarial algorithm here. Therefore, we introduce a new problem of Lipschitz bandits under the *switching* environment, and propose the Zooming TS algorithm with a restart trick to deal with the "almost stationary" nature of the bandit hyperparameter tuning problem.

## 3 Preliminaries

We first review the problem setting of contextual bandit algorithms. Denote $T$ as the total number of rounds and $K$ as the number of arms we could choose at each round, where $K$ could be infinite. At each round $t \in [T] := \{1, \ldots, T\}$, the player is given $K$ arms represented by a set of feature vectors $\mathcal{X}_t = \{x_{t,a} \mid a \in [K]\} \subseteq \mathbb{R}^d$ drawn from some unknown distribution, where $x_{t,a}$ is a $d$-dimensional vector containing information of arm $a$ at round $t$. The player selects an action $a_t \in [K]$ based on the current $\mathcal{X}_t$ and previous observations, and only receives the payoff of the pulled arm $a_t$. Denote $x_t := x_{t,a_t}$ as the feature vector of the chosen arm $a_t$ and $y_t$ as the corresponding reward. We assume the reward $y_t$ follows a canonical exponential family with minimal representation, a.k.a. generalized linear bandits (GLB) with some mean function $\mu(\cdot)$. In addition, one can represent this model by $y_t = \mu(x_t^\top \theta^*) + \epsilon_t$, where $\epsilon_t$ follows a sub-Gaussian distribution with parameter $\sigma^2$ independent with the information filtration $\mathcal{F}_t = \sigma(\{a_s, \mathcal{X}_s, y_s\}_{s=1}^{t-1})$ and $\sigma(\mathcal{X}_t)$ up to round $t$, and $\theta^*$ is some unknown coefficient. Denote $a_{t,*} := \arg\max_{a \in [K]} \mu(x_{t,a}^\top \theta^*)$ as the optimal arm at round $t$ and $x_{t,*}$ as its corresponding feature vector. The goal is to minimize the expected cumulative regret defined as:

$$R(T) = \sum_{t=1}^{T} \left[ \mu(x_{t,*}^\top \theta^*) - \mathbb{E}\left(\mu(x_t^\top \theta^*)\right) \right]. \tag{1}$$

Note that all state-of-the-art contextual GLB algorithms depend on at least one hyperparameter to balance the well-known exploration-exploitation tradeoff. For example, LinUCB (Li et al., 2010), the most popular UCB linear bandit, uses the following rule for arm selection at round $t$:

$$a_t = \arg\max_{a \in [K]} x_{t,a}^\top \hat{\theta}_t + \alpha_1(t) \|x_{t,a}\|_{V_t^{-1}}. \tag{LinUCB}$$

Here the model parameter $\hat{\theta}_t$ is estimated at each round $t$ via ridge regression, i.e. $\hat{\theta}_t = V_t^{-1} \sum_{s=1}^{t-1} x_s y_s$ where $V_t = \lambda I_r + \sum_{s=1}^{t-1} x_s x_s^\top$. And it considers the standard deviation of each arm with an exploration parameter $\alpha_1(t)$, where with a larger value of $\alpha_1(t)$ the algorithm will be more likely to explore uncertain arms. Note that the regularization parameter $\lambda$ is only used to ensure $V_t$ is invertible and hence its value is not crucial and commonly set to 1. In theory we can choose the value of $\alpha_1(t)$ as $\alpha_1(t) = \sigma \sqrt{r \log\left((1 + t/\lambda)/\delta\right)} + \|\theta^*\| \sqrt{\lambda}$, to achieve the optimal $\widetilde{O}(\sqrt{T})$ bound of regret: However, in practice, the values of $\sigma$ and $\|\theta^*\|$ are unspecified, and hence this theoretical value of $\alpha_1(t)$ is inaccessible. Furthermore, it has been shown that this is a very conservative choice that would lead to unsatisfactory practical performance, and the practically optimal hyperparameter values to use are distinct and far from the theoretical ones under different algorithms or settings. We also conduct a series of simulations with several state-of-the-art GLB algorithms to validate this fact, which is deferred to Appendix A.1. Conclusively, the best exploration parameter to use in practice should always be chosen dynamically based on the specific scenario and past observations. In addition, many GLB algorithms depend on some other hyperparameters, which may also affect the performance. For example, SGD-TS also involves a stepsize parameter for the stochastic gradient descent besides the exploration rate, and it is well known that a decent stepsize could remarkably accelerate the convergence (Loizou et al., 2021). To handle all these cases, we propose a general framework that can be used to automatically tune multiple continuous hyperparameters for a contextual bandit.

For a certain contextual bandit, assume there are $p$ different hyperparameters $\alpha(t) = \{\alpha_i(t)\}_{i=1}^p$, and each hyperparameter $\alpha_i(t)$ could take values in an interval $[a_i, b_i]$, $\forall t$. Denote the parameter space $A = \bigotimes_{i=1}^p [a_i, b_i]$, and the theoretical optimal values as $\alpha^*(t)$. Given the observations $\mathcal{F}_t$ up to round $t$, we write $a_t(\alpha(t)|\mathcal{F}_t)$ as the arm we pulled when the hyperparameters are set to $\alpha(t)$, and $x_t(\alpha(t)|\mathcal{F}_t)$ as the corresponding feature vector.

Motivated by the success of Bayesian optimization (Frazier, 2018) on the hyperparameter tuning of the offline machine learning algorithms, the main idea of our algorithm is to formulate the hyperparameter optimization

---

**Algorithm 1** Zooming TS algorithm with Restarts

---

**Input:** Time horizon $T$, space $A$, epoch size $H$.

1: **for** $t = 1$ **to** $T$ **do**
2:      **if** $t \in \{\tau H + 1 : \tau = 0, 1, \dots\}$ **then**
3:          Initialize the total candidate space $A_0 = A$ and the active set $J \subseteq A_0$ s.t. $A_0 \subseteq \cup_{v \in J} B(v, r_1(v))$ and $n_1(v) \leftarrow 1, \forall v \in J$.      $\triangleright$Restart
4:      **else if** $\hat{f}_t(v) - \hat{f}_t(u) > r_t(v) + 2r_t(u)$ for some pair of $u, v \in J$ **then**
5:          Set $J = J \backslash \{u\}$ and $A_0 = A_0 \backslash B(u, r_t(u))$.      $\triangleright$Removal
6:      **end if**
7:      **if** $A_0 \not\subseteq \cup_{v \in J} B(v, r_t(v))$ **then**      $\triangleright$Activation
8:          Activate and pull some point $v \in A_0$ that has not been covered: $J = J \cup \{v\}, v_t = v$.
9:      **else**
10:          $v_t = \arg\max_{v \in J} I_t(v)$, break ties arbitrarily.      $\triangleright$Selection
11:      **end if**
12:      Observe the reward $\tilde{y}_{t+1}$, and then update components in the Zooming TS algorithm: $n_{t+1}(v), \hat{f}_{t+1}(v), r_{t+1}(v), s_{t+1}(v)$ for the chosen $v_t \in J$:

$$n_{t+1}(v_t) = n_t(v_t) + 1, \quad \hat{f}_{t+1}(v_t) = (\hat{f}_t(v_t) n_t(v_t) + \tilde{y}_{t+1}) / n_{t+1}(v_t).$$

13: **end for**

---

as a (another layer of) non-stationary Lipschitz bandit in the continuous space $A \subseteq \mathbb{R}^p$, i.e. the agent chooses an arm (hyperparameter combination) $\alpha \in A$ in round $t \in [T]$, and then we decompose $\mu(x_t(\alpha | \mathcal{F}_t)^\top \theta^*)$ as

$$\mu(x_t(\alpha | \mathcal{F}_t)^\top \theta^*) = g_t(\alpha) + \eta_{\mathcal{F}_t, \alpha}. \tag{2}$$

Here $g_t$ is some time-dependent Lipschitz function that formulates the performance of the bandit algorithm under the hyperparameter combination $\alpha$ at round $t$, since the bandit algorithm tends to pull similar arms if the chosen values of hyperparameters are close at round $t$. In other words, we expect close hyperparameter values to yield similar results with other conditions fixed, as in Bayesian optimization on offline hyperparameter tuning. To demonstrate that our Lipschitz assumption w.r.t. the hyperparameter values in Eqn. equation 3 is reasonable, we conduct simulations with LinUCB and LinTS, and defer it to Appendix A due to the space limit. Moreover, $(\eta_{\mathcal{F}_t, \alpha} - \mathbb{E}[\eta_{\mathcal{F}_t, \alpha}])$ is IID sub-Gaussian with parameter $\tau^2$, and to be fair we assume $\mathbb{E}[\eta_{\mathcal{F}_t, \alpha}]$ could also depend on the history $\mathcal{F}_t$ since past observations and action sets would explicitly influence the model parameter estimation and hence the decision making at each round. In addition to Lipschitzness, we also suppose $g_t$ follows a *switching* environment: $g_t$ is piecewise stationary with some change points, i.e.

$$|g_t(\alpha_1) - g_t(\alpha_2)| \leq \|\alpha_1 - \alpha_2\|, \ \forall \alpha_1, \alpha_2 \in A; \tag{3}$$

$$\sum_{t=1}^{T-1} \mathbf{1}[\exists \alpha \in A : g_t(\alpha) \neq g_{t+1}(\alpha)] = c(T), \ c(T) \in \mathbb{N}. \tag{4}$$

Since after sufficient exploration, the expected reward should be stable with the same hyperparameter setting, we could assume that $c(T) = \tilde{O}(1)$. Detailed justification on this piecewise Lipschitz assumption is deferred to Remark B.1 in Appendix B. Although numerous research works have considered the *switching* environment (a.k.a. *abruptly-changing* environment) for multi-armed or linear bandits (Auer et al., 2002b; Wei et al., 2016), our work is the first to introduce this setting into the continuum-armed bandits. In Section 4.1, we will show that by combining our proposed Zooming TS algorithm for Lipschitz bandits with a simple restarted strategy, a decent regret bound could be achieved under the *switching* environment.

## 4 Main Results

In this section, we present our novel online hyperparameter optimization framework that could be easily adapted to most contextual bandit algorithms. We first introduce the continuum-arm Lipschitz bandit problem under the *switching* environment, and propose the Zooming TS algorithm with Restarts which

Figure 1: Illustration of the restarted strategy.

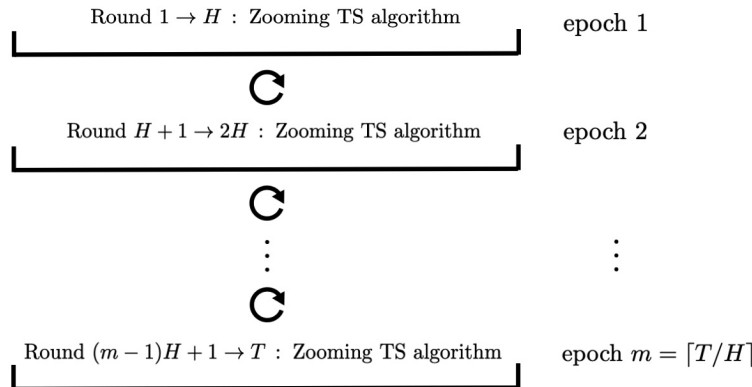

modifies the traditional Zooming algorithm (Kleinberg et al., 2019) to make it more efficient and also adaptive to the *switching* environment. Subsequently, we propose our bandit hyperparameter tuning framework named Continuous Dynamic Tuning (CDT) by making use of our proposed Zooming TS algorithm with Restarts and the Bandit-over-Bandit (BOB) idea.

W.l.o.g. we assume that there exists a positive constant $S$ such that $\|\theta^*\| \leq S$ and $\|x_{t,a}\| \leq 1$, $\forall t, a$, and each hyperparameter space has been shifted and scaled to $[0, 1]$. We also assume that the mean reward $\mu(x_{t,a}^\top \theta^*) \in [0, 1]$, and hence naturally $g_t(\alpha) \in [0, 1]$, $\forall \alpha \in A = [0, 1]^p, t \in [T]$.

### 4.1 Zooming TS Algorithm with Restarts

For simplicity and consistency, we will reload and introduce a new system of notations in this subsection. Consider the non-stationary Lipschitz bandit problem on a compact space $A$ under some metric $\text{Dist}(\cdot, \cdot) \geq 0$, where the covering dimension is denoted by $p_c$. The learner pulls an arm $v_t \in A$ at round $t \in [T]$ and subsequently receives a reward $\tilde{y}_t$ sampled independently of $\mathbb{P}_{v_t}$ as $\tilde{y}_t = f_t(v_t) + \eta_v$, where $t = 1, \ldots, T$ and $\eta_v$ is IID zero-mean error with sub-Guassian parameter $\tau_0^2$, and $f_t$ is the expected reward function at round $t$ and is Lipschitz with respect to $\text{Dist}(\cdot, \cdot)$. The *switching* environment assumes the time horizon $T$ is partitioned into $c(T) + 1$ intervals, and the bandit stays stationary within each interval, i.e.

$$|f_t(m) - f_t(n)| \leq \text{Dist}(m, n), \ \ m, n \in A; \text{ and } \sum_{t=1}^{T-1} \mathbf{1}[\exists m \in A : f_t(m) \neq f_{t+1}(m)] = c(T).$$

Here in this section $c(T) = o(T)$ could be any integer. The goal of the learner is to minimize the expected (dynamic) regret that is defined as:

$$R_L(T) = \sum_{t=1}^{T} \max_{v \in A} f_t(v) - \sum_{t=1}^{T} \mathbb{E}\left(f_t(v_t)\right).$$

At each round $t$, $v_t^* \coloneqq \arg\max_{v \in A} f_t(v)$ denotes the maximal point (w.l.o.g. assume it's unique), and $\Delta_t(v) = f_t(v^*) - f_t(v)$ is the "badness" of each arm $v$. We also denote $A_{r,t}$ as the $r$-optimal region at the scale $r \in (0, 1]$, i.e. $A_{r,t} = \{v \in A : r/2 < \Delta_t(v) \leq r\}$ at time $t$. Then the $r$-zooming number $N_{z,t}(r)$ of $(A, f_t)$ is defined as the minimal number of balls of radius no more than $r$ required to cover $A_{r,t}$. (Note the subscript $z$ stands for zooming here.) Next, we define the zooming dimension $p_{z,t}$ (Kleinberg et al., 2019) at time $t$ as the smallest $q \geq 0$ such that for every $r \in (0, 1]$ the $r$-zooming number can be upper bounded by $cr^{-q}$ for some multiplier $c > 0$ free of $r$:

$$p_{z,t} = \min\{q \geq 0 : \exists c > 0, N_{z,t}(r) \leq cr^{-q}, \forall r \in (0, 1]\}.$$

It's obvious that $0 \leq p_{z,t} \leq p_c$, $\forall t \in [T]$. (Note $p_{z,t}$ is fixed under the stationary environment.) On the other hand, the zooming dimension could be much smaller than $p_c$ under some mild conditions. For example, if

the payoff function $f_t$ defined on $\mathbb{R}^{p_c}$ is greater than $\|v_t^* - v\|^\beta$ in scale for some $\beta \geq 1$ around $v^*$ in the space $A$, i.e. $f_t(v_t^*) - f_t(v) = \Omega(\|v_t^* - v\|^\beta)$, then it holds that $p_{z,t} \leq (1 - 1/\beta)p_c$. Note that we have $\beta = 2$ (i.e. $p_{z,t} \leq p_c/2$) when $f_t(\cdot)$ is $C^2$-smooth and strongly concave in a neighborhood of $v^*$. More details are presented in Appendix C. Since the expected reward Lipschitz function $f_t(\cdot)$ is fixed in each time interval under the *switching* environment, the zooming number and zooming dimension $p_{z,t}$ would also stay identical. And we also write $p_{z,*} = \max_{t\in[T]} p_{z,t} \leq p_c$.

Our proposed Algorithm 1 extends the classic Zooming algorithm (Kleinberg et al., 2019), which was used under the stationary Lipschitz bandit environment, by adding several new ingredients for better efficiency and adaptivity to non-stationary environment: on the one hand, we employ the TS methodology and propose a novel removal step. Here we utilize TS since it was shown that TS is more robust than UCB in practice (Chapelle & Li, 2011; Wang & Chen, 2018), and the removal procedure in line 5 of Algorithm 1 could adaptively subtract regions that are prone to yield low rewards. Both of these two ideas could enhance the algorithmic efficiency, which coincides with the practical orientation of our work. On the other hand, the restarted strategy proceeds our proposed Zooming TS in epochs and refreshes the algorithm after every $H$ rounds, as displayed in Figure 1. The epoch size $H$ is fixed through the total time horizon and controls the tradeoff between non-stationarity and stability. Note that $H$ in our algorithm does not need to match the actual length of stationary intervals of the environment, and we would discuss its selection later. At each epoch, we maintain a time-varying active arm set $S_t \subseteq A$, which is initially empty and updated every time. For each arm $v \in A$ and time $t$, denote $n_t(v)$ as the number of times arm $v$ has been played before time $t$ since the last restart, and $\hat{f}_t(v)$ as the corresponding average sample reward. We let $\hat{f}_t(v) = 0$ when $n_t(v) = 0$. Define the confidence radius and the TS standard deviation of active arm $v$ at time $t$ respectively as

$$r_t(v) = \sqrt{\frac{13\tau_0^2 \ln T}{2n_t(v)}}, \quad s_t(v) = s_0\sqrt{\frac{1}{n_t(v)}}, \tag{5}$$

where $s_0 = \sqrt{52\pi\tau_0^2 \ln(T)}$. We call $B(v, r_t(v)) = \{u \in \mathbb{R}^p : \text{Dist}(u, v) \leq r_t(v)\}$ as the confidence ball of arm $v$ at time $t \in [T]$. We construct a randomized algorithm by choosing the best active arm according to the perturbed estimate mean $I_t(\cdot)$:

$$I_t(v) = \hat{f}_t(v) + s_t(v)Z_{t,v}, \tag{6}$$

where $Z_{t,v}$ is i.i.d. drawn from the clipped standard normal distribution: we first sample $\tilde{Z}_{t,v}$ from the standard normal distribution and then set $Z_{t,v} = \max\{1/\sqrt{2\pi}, \tilde{Z}_{t,v}\}$. This truncation was also used in TS multi-armed bandits (Jin et al., 2021), and our algorithm clips the posterior samples with a lower threshold to avoid underestimation of good arms. Moreover, the explanations of the TS update is deferred to Appendix D due to the space limit.

The regret analysis of Algorithm 1 is very challenging since the active arm set is constantly changing and the optimal arm $v^*$ cannot be exactly recovered under the Lipschitz bandit setting. Thus, existing theory on multi-armed bandits with TS is not applicable here. We overcome these difficulties with some innovative use of metric entropy theory, and the regret bound of Algorithm 1 is given as follows.

**Theorem 4.1.** *With $H = \Theta\left((T/c(T))^{(p_{z,*}+2)/(p_{z,*}+3)}\right)$, the total regret of our Zooming TS algorithm with Restarts under the switching environment over time $T$ is bounded as*

$$R_L(T) \leq \tilde{O}\left((c(T))^{1/(p_{z,*}+3)} T^{(p_{z,*}+2)/(p_{z,*}+3)}\right),$$

*when $c(T) > 0$. In addition, if the environment is stationary (i.e. $c(T) = 0, f_t = f, p_{z,t} = p_{z,*} := p_z, \forall t \in [T]$), then by using $H = T$ (i.e. no restart), our Zooming TS algorithm could achieve the optimal regret bound for Lipschitz bandits up to logarithmic factors:*

$$R_L(T) \leq \tilde{O}\left(T^{(p_z+1)/(p_z+2)}\right).$$

We also present empirical studies to further evaluate the performance of our Algorithm 1 compared with stochastic Lipschitz bandit algorithms in Appendix A.3. A potential drawback of Theorem 4.1 is that

---

**Algorithm 2** Continuous Dynamic Tuning (CDT)

---

**Input:** $T_1, T_2, \{\mathcal{X}_t\}_{t=1}^T, A = \bigotimes_{i=1}^p [a_i, b_i].$
1: Randomly choose $a_t \in [K]$ and observe $x_t, y_t, t \leq T_1$.
2: Initialize the hyperparameter active set $J$ s.t. $A \subseteq \cup_{v \in J} B(v, r_1(v))$ where $n_{T_1}(v) \leftarrow 1, \forall v \in J$.
3: **for** $t = (T_1 + 1)$ **to** $T$ **do**
4:     Run the $t$-th iteration of Algorithm 1 with initial input horizon $T - T_1$, input space $A$ and restarting epoch length $T_2$. Denote the pulled arm at round $t$ as $\alpha(i_t) \in A$.       ▷Top
5:     Run the contextual bandit algorithm with hyperparameter $\alpha(i_t)$ to pull an arm $a_t$.     ▷Bottom
6:     Obtain $y_t$ and update components in the contextual bandit algorithm.     ▷Bottom Update
7:     Update components in Algorithm 1 by treating $y_t$ as the reward of arm $\alpha(i_t)$     ▷Top Update
8: **end for**

---

the optimal epoch size $H$ under *switching* environment relies on the value of $c(T)$ and $p_{z,*}$, which are unspecified in reality. However, this problem could be solved in theory by using the BOB idea (Cheung et al., 2019; Zhao et al., 2020) to adaptively choose the optimal epoch size with a meta algorithm (e.g. EXP3 (Auer et al., 2002b)) in real time. In this case, we prove the expected regret can be bounded by the order of $\tilde{O}\left(T^{\frac{p_c+2}{p_c+3}} \cdot \max\left\{c(T)^{\frac{1}{p_c+3}}, T^{\frac{1}{(p_c+3)(p_c+4)}}\right\}\right)$ in general, and some better regret bounds in problem-dependent cases. More details are presented in Theorem F.1 with its proof in Appendix F. However, in the following Section 4.2 we could simply set $H = T^{(2+p)/(3+p)}$ in our CDT framework where $p$ is the number of hyperparameters to be tuned after assuming $c(T) = \tilde{O}(1)$ is of constant scale up to logarithmic terms. The value of $\tau_0$ can be determined by assuring the observed rewards are bounded. Note our work introduces a new problem on Lipschitz bandits under the *switching* environment. One potential limitation of our work is how to deduce a regret lower bound under this problem setting is unclear, and we leave it as a future work.

## 4.2 Online Continuous Hyperparameter Optimization for Contextual Bandits

Based on the proposed algorithm in the previous subsection, we introduce our online double-layer Continuous Dynamic Tuning (CDT) framework for hyperparameter optimization of contextual bandit algorithms. We assume the arm to be pulled follows a fixed distribution given the hyperparameters to be used and the history at each round. The detailed algorithm is shown in Algorithm 2. Our method extends the bandit-over-bandit (BOB) idea that was first proposed for non-stationary stochastic bandit problems (Cheung et al., 2019), where it adjusts the sliding-window size dynamically based on the changing model. In our work, for the top layer we use our proposed Algorithm 1 to tune the best hyperparameter values from the admissible space, where each arm represents a hyperparameter configuration and the corresponding reward is the algorithmic result. $T_2$ is the length of each epoch (i.e. $H$ in Algorithm 1), and we would refresh our Zooming TS Lipschitz bandit after every $T_2$ rounds as shown in Line 5 of Algorithm 2 due to the non-stationarity. The bottom layer is the primary contextual bandit and would run with the hyperparameter values $\alpha(i_t)$ chosen from the top layer at each round $t$. We also include a warming-up period of length $T_1$ in the beginning to guarantee sufficient exploration as in Li et al. (2017); Ding et al. (2021). Despite the focus of our CDT framework is on the practical aspect, we also present a novel theoretical analysis in the following for the completeness of our work.

Although there has been a rich line of work on regret analysis of UCB and TS GLB algorithms, most literature certainly requires that some hyperparameters, e.g. exploration rate, always take their theoretical values. It is challenging to study the regret bound of GLB algorithms when their hyperparameters are synchronously tuned in real time, since the chosen hyperparameter values may be far from the theoretical ones in practice, not to mention that previous decisions would also affect the current update cumulatively. Moreover, there is currently no existing literature and regret analysis on hyperparameter tuning (or model selection) for bandit algorithms with an infinite number of candidates in a continuous space. Recall that we denote $\mathcal{F}_t = \sigma\left(\{a_s, \mathcal{X}_s, y_s\}_{s=1}^{t-1}\right)$ as the past information before round $t$ under our CDT framework, and $a_t, x_t$ are the chosen arm and its corresponding feature vector at time $t$, which implies that $a_t = a_t(\alpha(i_t)|\mathcal{F}_t), x_t = x_t(\alpha(i_t)|\mathcal{F}_t)$. Furthermore, we denote $\alpha^*(t)$ as the theoretical optimal value at round $t$ and $\mathcal{F}_t^*$ as the past information filtration by always using the theoretical optimal $\alpha^*(t)$. Since the decision at each round $t$ also depends on the history observe by time $t$, the pulled arm with the same hyperparameter $\alpha(t)$ might be different under $\mathcal{F}_t$ or $\mathcal{F}_t^*$. To

analyze the cumulative regret $R(T)$ of our Algorithm 2, we first decompose it into four quantities:

$$R(T) = \underbrace{\mathbb{E}\left[\sum_{t=1}^{T_1} \left(\mu(x_{t,*}^\top\theta^*) - \mu(x_t^\top\theta^*)\right)\right]}_{\text{Quantity (A)}} + \underbrace{\mathbb{E}\left[\sum_{t=T_1+1}^{T} \left(\mu(x_{t,*}^\top\theta^*) - \mu(x_t(\alpha^*(t)|\mathcal{F}_t^*)^\top\theta^*)\right)\right]}_{\text{Quantity (B)}}$$

$$+ \underbrace{\mathbb{E}\left[\sum_{t=T_1+1}^{T} \left(\mu\left(x_t(\alpha^*(t)|\mathcal{F}_t^*)^\top\theta^*\right) - \mu(x_t(\alpha^*(t)|\mathcal{F}_t)^\top\theta^*)\right)\right]}_{\text{Quantity (C)}}$$

$$+ \underbrace{\mathbb{E}\left[\sum_{t=T_1+1}^{T} \left(\mu\left(x_t(\alpha^*(t)|\mathcal{F}_t)^\top\theta^*\right) - \mu(x_t(\alpha(i_t)|\mathcal{F}_t)^\top\theta^*)\right)\right]}_{\text{Quantity (D)}}.$$

Intuitively, Quantity (A) is the regret paid for pure exploration during the warming-up period and could be controlled by the order $O(T_1)$. Quantity (B) is the regret of the contextual bandit algorithm that runs with the theoretical optimal hyperparameters $\alpha^*(t)$ all the time, and hence it could be easily bounded by the optimal scale $\tilde{O}(\sqrt{T})$ based on the literature. Quantity (C) is the difference of cumulative reward with the same $\alpha^*(t)$ under two separate lines of history. Quantity (D) is the extra regret paid to tune the hyperparameters on the fly. By using the same line of history $\mathcal{F}_t$ in Quantity (D), the regret of our Zooming TS algorithm with Restarts in Theorem 4.1 can be directly used to bound Quantity (D). Conclusively, we deduce the following theorem for the regret bound:

**Theorem 4.2.** *Under our problem setting in Section 3, for UCB and TS GLB algorithms with exploration hyperparameters (e.g. LinUCB, UCB-GLM, GLM-UCB, LinTS), by taking $T_1 = O(T^{2/(p+3)}), T_2 = O(T^{(p+2)/(p+3)})$ where $p$ is the number of hyperparameters, and let the theoretically optimal hyperparameter combination $\alpha^*(T) \in A$, it holds that*

$$\mathbb{E}[R(T)] \leq \tilde{O}(T^{(p+2)/(p+3)}).$$

The detailed proof of Theorem 4.2 is presented in Appendix G. Note that this regret bound could be further improved to $\tilde{O}(T^{(p_0+2)/(p_0+3)})$ where $p_0$ is any constant that is no smaller than the zooming dimension of $(A, g_t), \forall t$. For example, from Figure 3 in Appendix A we can observe that in practice $g_t$ would be $C^2$-smooth and strongly concave, which implies that $\mathbb{E}[R(T)] \leq \tilde{O}(T^{(p+4)/(p+6)})$.

Note our work is the first one to consider model selection for bandits with a continuous candidate set, and the regret analysis for online model selection in the bandit setting (Foster et al., 2019) is intrinsically difficult. For example, regret bounds of the algorithm CORRAL (Agarwal et al., 2017) for model selection and Syndicated (Ding et al., 2022b) for bandit hyperparameter tuning are (sub)linearly dependent on the number of candidates, which would be infinitely large and futile in our case. Furthermore, given the fact that Syndicated in Ding et al. (2022b) fails to recover the optimal $O(\sqrt{T})$ bound of regret without stringent assumptions under the easier setting with finite hyperparameter candidates, it would be substantially difficult to deduce a feasible regret bound under our more complicated problem setting. Moreover, the non-stationarity under the *switching* environment would further deteriorate the optimal order of cumulative regret Cheung et al. (2019). And it is intrinsically more difficult to consider the continuum-armed bandit over the multi-armed bandit. Therefore, we believe our theoretical result is non-trivial and significant. Our work stands as the first seminal attempt in bandit hyperparameter tuning (or even bandit model selection) with an infinite number of candidates. An extensive study on this new problem will be an interesting future direction.

## 5 Experimental Results

In this section, we show by experiments that our hyperparameter tuning framework outperforms the theoretical hyperparameter setting and other tuning methods with various (generalized) linear bandit algorithms. We

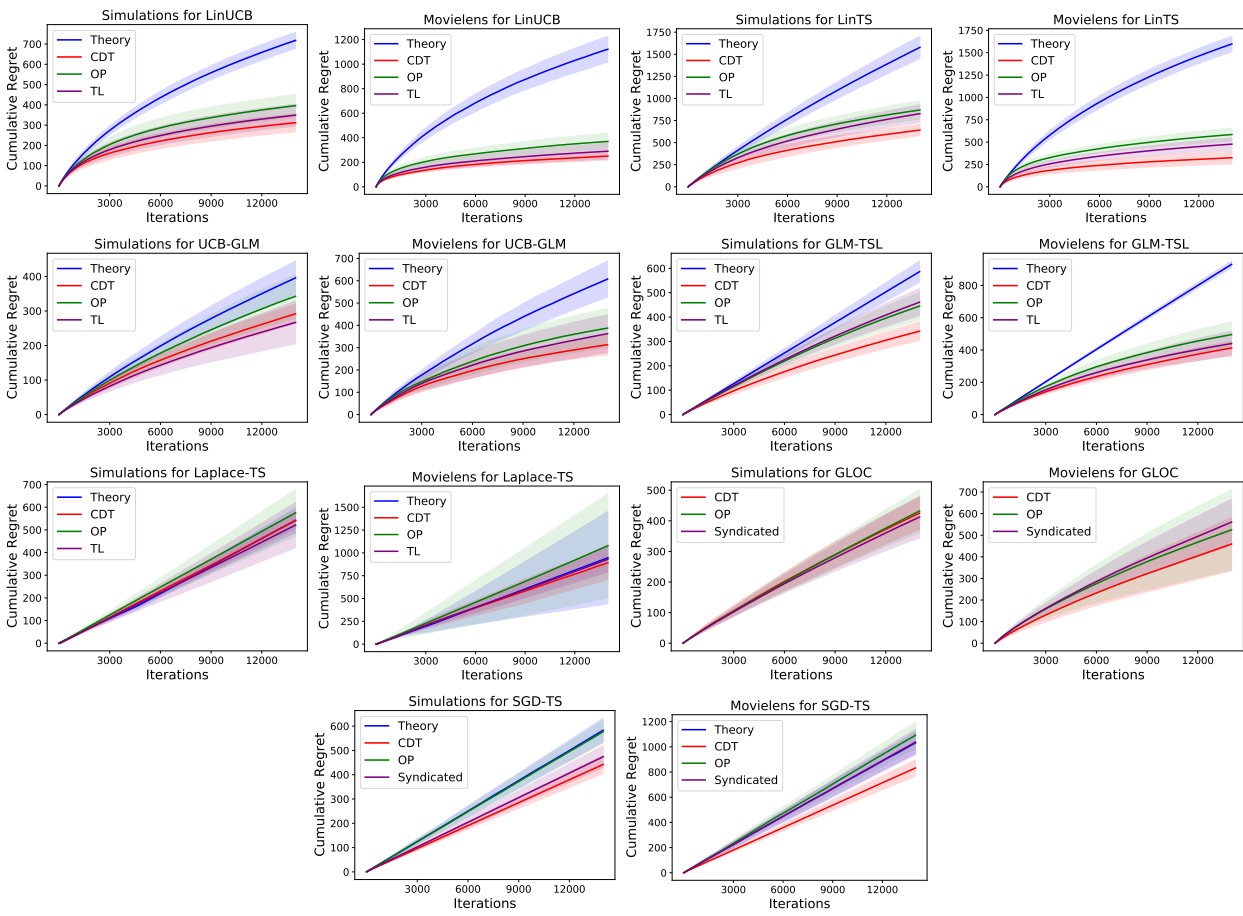

Figure 2: Cumulative regret curves of our CDT framework compared with existing hyperparameter selection methods under multiple (generalized) linear bandit algorithms on the simulations and Movielens dataset.

utilize seven state-of-the-art bandit algorithms: two of them (LinUCB (Li et al., 2010), LinTS (Agrawal & Goyal, 2013)) are linear bandits, and the other five algorithms (UCB-GLM (Li et al., 2017), GLM-TSL (Kveton et al., 2020), Laplace-TS (Chapelle & Li, 2011), GLOC (Jun et al., 2017), SGD-TS (Ding et al., 2021)) are GLBs. Note that all these bandit algorithms except Laplace-TS contain an exploration rate hyperparameter, while GLOC and SGD-TS further require an additional learning parameter. And Laplace-TS only depends on one stepsize hyperparameter for a gradient descent optimizer. We compare our CDT framework with the theoretical setting, OP (Bouneffouf & Claeys, 2020) and TL (Ding et al., 2022b) (one hyperparameter) and Syndicated (Ding et al., 2022b) (multiple hyperparameters) algorithms. Their details are given as follows:

(1) **Theoretical setting**: We implement the theoretical exploration rate and stepsize for each algorithm. For the stepsize of gradient descent used in SGD-TS and Laplace-TS, we set it as 1 instead. (We observe the algorithmic performance is not sensitive to this stepsize.)

(2) **OP**: (Bouneffouf & Claeys, 2020) proposes OPLINUCB to tune the exploration rate of LinUCB. Here we modify it so that it could be used in other bandit algorithms. Note that OP is only applicable to algorithms with one hyperparameter, and hence we fix the learning parameter of GLOC and SGD-TS as their theoretical values instead, and only tune the exploration rates.

(3) **TL** (Ding et al., 2022b) (one hyperparameter): For algorithms with only one hyperparameter, TL is used.

(4) **Syndicated** (Ding et al., 2022b) (multiple hyperparameters): For GLOC and SGD-TS (two hyperparameters), the Syndicated framework is utilized for comparison.

We run comprehensive experiments on both simulations and real-world datasets. Specifically, for the real data, we use the benchmark Movielens 100K dataset along with the Yahoo News dataset:

Table 1: Running time (seconds) for different algorithms under settings shown in Figure 2.

| Algorithm | Setting | Theory | TL | OP | CDT |
|---|---|---|---|---|---|
| LinUCB | Simulation | 2.11 | 4.01 | 3.70 | 6.89 |
| | Movielens | 2.17 | 3.87 | 2.95 | 7.31 |
| LinTS | Simulation | 2.21 | 4.10 | 3.95 | 7.63 |
| | Movielens | 2.04 | 4.09 | 3.45 | 7.71 |
| UCB-GLM | Simulation | 7.74 | 9.84 | 9.71 | 12.67 |
| | Movielens | 7.89 | 9.64 | 9.35 | 13.05 |
| GLM-TSL | Simulation | 304.28 | 306.98 | 306.02 | 309.54 |
| | Movielens | 294.17 | 295.87 | 294.83 | 298.81 |
| Laplace-TS | Simulation | 523.62 | 526.31 | 526.24 | 531.15 |
| | Movielens | 500.19 | 503.45 | 503.91 | 509.65 |
| GLOC | Simulation | 486.58 | 489.31 | 490.24 | 593.15 |
| | Movielens | 474.87 | 476.52 | 477.16 | 481.03 |
| SGD-TS | Simulation | 67.42 | 70.62 | 69.42 | 73.68 |
| | Movielens | 62.09 | 66.48 | 64.68 | 67.31 |

(1) **Simulation**: In each repetition, we simulate all the feature vectors $\{x_{t,a}\}$ and the model parameter $\theta^*$ according to Uniform$(-1/\sqrt{r}, 1/\sqrt{r})$ elementwisely, and hence we have $\|x_{t,a}\| \leq 1$. We set $d = 25$, $K = 120$ and $T = 14,000$. For linear model, the expected reward of arm $a$ is formulated as $x_{t,a}^\top \theta^*$ and random noise is sampled from $N(0, 0.25)$; for Logistic model, the mean reward of arm $a$ is defined as $p = 1/(1 + \exp(-x_{t,a}^\top \theta^*))$, and the output is drawn from a Bernoulli distribution.

(2) **Movielens 100K dataset**: This dataset contains 100K ratings from 943 users on 1,682 movies. For data pre-processing, we utilize LIBPMF (Yu et al., 2014) to perform matrix factorization and obtain the feature matrices for both users and movies with $d = 20$, and then normalize all feature vectors into unit $r$-dimensional ball. In each repetition, the model parameter $\theta^*$ is defined as the average of 300 randomly chosen users' feature vectors. And for each time $t$, we randomly choose $K = 300$ movies from 1,682 available feature vectors as arms $\{x_{t,a}\}_{a=1}^{300}$. The time horizon $T$ is set to 14,000. For linear models, the expected reward of arm $a$ is formulated as $x_{t,a}^\top \theta^*$ and random noise is sampled from $N(0, 0.5)$; for Logistic model, the output of arm $a$ is drawn from the Bernoulli distribution with $p = 1/(1 + \exp(-x_{t,a}^\top \theta^*))$.

(3) **Yahoo News dataset:** We downloaded the Yahoo Recommendation dataset R6A, which contains Yahoo data from May 1 to May 10, 2009 with $T = 2881$ timestamps. For each user's visit, the module will select one article from a pool of 20 articles for the user, and then the user will decide whether to click. We transform the contextual information into a 6-dimensional vector based on the processing in (Chu et al., 2009). We build a Logistic bandit on this data, and the observed reward is simulated from a Bernoulli distribution with a probability of success equal to its click-through rate at each time.

We first present the results on simulations and Movielens datasets: since all the existing tuning algorithms require a user-defined candidate set, we design the tuning set for all potential hyperparameters as $\{0.1, 1, 2, 3, 4, 5\}$. And for our CDT framework, which is the first algorithm for tuning hyperparameters in an interval, we simply set the interval as $[0.1, 5]$ for all hyperparameters. Each experiment is repeated for 20 times, and the average regret curves with standard deviation are displayed in Figure 2. We further explore the existing methods after enlarging the hyperparameter candidate set to fairly validate the superiority of our proposed CDT in Appendix A.4.1. The results in Appendix A.4.1 further lead to discussion on why it is inefficient to first discretize the continuous space and then implement an algorithm (e.g. Syndicated) with discrete candidate sets.

We believe a large value of warm-up period $T_1$ may abandon some useful information in practice, and hence we use $T_1 = T^{2/(p+3)}$ according to Theorem 4.2 in experiments. And we would restart our hyperparameter tuning layer after every $T_2 = 3T^{(p+2)/(p+3)}$ rounds. An ablation study on the role of $T_1, T_2$ in our CDT framework is also conducted and deferred to Appendix A.4.2, where we demonstrate that the performance of CDT is pretty robust to the choice of $T_1, T_2$ in practice.

From Figure 2, we observe that our CDT framework outperforms all existing hyperparameter tuning methods for most contextual bandit algorithms. It is also clear that CDT performs stably and soundly with the smallest standard deviation across most datasets (e.g. experiments for LinTS, UCB-GLM), indicating that our method is highly flexible and robustly adaptive to different datasets. Moreover, when tuning multiple hyperparameters (GLOC, SGD-TS), we can see that the advantage of our CDT is also evident since our method is intrinsically designed for any hyperparameter space. It is also verified that the theoretical hyperparameter values are too conservative and would lead to terrible performance (e.g. LinUCB, LinTS). Note that all tuning methods exhibit similar results when applied to Laplace-TS. We believe it is because Laplace-TS only relies on an insensitive hyperparameter that controls the stepsize in gradient descent loops, which mostly affects the convergence speed. To further validate the high efficiency of our proposed CDT, we also report the computational running time for the 14 cases corresponding to Figure 2. Specifically, we display the average running time on each method in Table 1. We can observe that all existing methods and our CDT can run very fast in practice, and our CDT is only slightly more expensive than TL and OP in computation (CDT only takes about four more seconds) since the procedure of removal, restarting and activation checks at each round would take some extra computation. In addition, we can conclude that the main computation time comes from the contextual bandit algorithm we want to tune on, as is shown that, e.g. GLM-TSL requires much more time than all other methods under different tuning methods. Therefore, we can conclude that our CDT significantly outperforms all existing baselines without increasing computational time.

For the Yahoo News Recommendation dataset, since it is a logistic bandit, we only output the cumulative rewards of GLBs in Table 2. From the table, we can observe that our proposed CDT also performs the best overall. Specifically, it is only slightly worse than TL for GLM-TSL and GLOC, and yields the best results among all hyperparameter tuning frameworks for UCB-GLM, GLM-TSL, and SGD-TS. And the theoretical hyperparameter setting is very unstable again as in Figure 2. Conclusively, our proposed CDT yields uniformly the best performances compared with existing baselines in both large-scale and mild-scale experiments with multiple contextual bandit algorithms. This fact also validates the rationality of Lipschitz continuity assumption on the bandit hyperparameter tuning problem in Section 3.

| Method | UCB-GLM | GLM-TSL | Laplace-TS | GLOC | SGD-TS |
|---|---|---|---|---|---|
| Theory | 221.51 | 214.67 | 217.38 | | 206.73 |
| CDT | 221.69 | 218.27 | 217.05 | 217.95 | 218.35 |
| OP | 217.25 | 217.08 | 213.95 | 216.28 | 215.58 |
| TL/Syndicated | 218.95 | 219.36 | 214.42 | 218.19 | 215.02 |

Table 2: Comparisons of cumulative rewards from different algorithms on Yahoo dataset.

# 6 Conclusion

In this paper, we propose the first online continuous hyperparameter optimization method for contextual bandit algorithms named CDT given the continuous hyperparameter search space. Our framework can attain sublinear regret bound in theory, and is general enough to handle the hyperparameter tuning task for most contextual bandit algorithms. Multiple synthetic and real experiments with multiple GLB algorithms validate the remarkable efficiency of our framework compared with existing methods in practice. In the meanwhile, we propose the Zooming TS algorithm with Restarts, which is the first work on Lipschitz bandits under the *switching* environment.

**Limitations and future works:** Beyond the hyperparameter selection, our work paves the way for exploring the broader problem of bandit model selection within a continuous candidate space. Another promising avenue for future investigation involves conducting a comprehensive study on the non-stationary Lipschitz bandit problem. The examination of lower bounds for these two novel directions falls outside the purview of our study but remains intriguing for further exploration.

## Acknowledgments

We appreciate the insightful comments from the TMLR reviewers and the action editor. This work was partially supported by the National Science Foundation under grants CCF-1934568, DMS-1916125, DMS-2113605, DMS-2210388, IIS-2008173 and IIS-2048280.

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

# A  Supportive Experimental Details

## A.1  Simulations on the Optimal Hyperparameter Value in Grid Search

To further validate the necessity of dynamic hyperparameter tuning, we conduct a simulation for UCB algorithms LinUCB, UCB-GLM, GLOC and TS algorithms LinTS, GLM-TSL with a grid search of exploration parameter in $\{0.1, 0.5, 1, 1.5, 2, \dots, 10\}$ and then report the best parameter value under different settings. Specifically, we set $d = 10, T = 8000, K = 60, 120$, and choose arm $x_{t,a}$ and $\theta^*$ randomly in $\{x : \|x\| \leq 1\}$. Rewards are simulated from $N(x_{t,a}^\top \theta^*, 0.5)$ for LinUCB, LinTS, and from Bernoulli$(1/(1 + \exp(-x_{t,a}^\top \theta^*)))$ for UCB-GLM, GLOC and GLM-TSL. The results are displayed in Table 3, where we can see that the optimal hyperparameter values are distinct and far from the theoretical ones under different algorithms or settings. Moreover, the theoretical optimal exploration rate should be identical under different values of $K$ for most algorithms shown here, but in practice the best hyperparameter to use depends on $K$, which also contradicts with the theoretical result.

| Bandit type | Linear bandit | | Generalized linear bandit | | |
|---|---|---|---|---|---|
| Algorithm | LinUCB | LinTS | UCB-GLM | GLOC | GLM-TSL |
| $K = 60$ | 2.5 | 1 | 1.5 | 4.5 | 1.5 |
| $K = 120$ | 3 | 1.5 | 2.5 | 5 | 2 |

Table 3: The optimal exploration parameter value in grid search for LinUCB, LinTS, UCB-GLM, GLOC and GLM-TSL based on average cumulative regret of 5 repeated simulations.

## A.2  Simulations to Validate the Lipschitzness of Hyperparameter Configuration

We also conduct another simulation to show it is reasonable and fair to assume the expected reward is an almost-stationary Lipschitz function w.r.t. hyperparameter values. Specifically, we set $d = 6, T = 3000, K = 60$, and for each time we run LinUCB and LinTS by using our CDT framework, but also obtain the results by choosing the exploration hyperparameter in the set $\{0.3, 0.45, 0.6, \dots, 8.85, 9\}$ respectively. For the first 200 rounds we use the random selection for sufficient exploration, and hence we omit the results for the first 200 rounds. After the warming-up period, we divide the rest of iterations into 140 groups uniformly, where each group contains 20 consecutive iterations. Then we calculate the mean of the obtained reward of each hyperparameter value in the adjacent 20 rounds, and centralize the mean reward across different hyperparameters in each group (we call it group mean reward). Afterward, we can calculate the mean and standard deviation of the group mean reward for different hyperparameter values across all groups. The results are shown in Figure 3, where we can see the group mean reward can be decently represented by a stationary Lipschitz continuous function w.r.t hyperparameter values. Conclusively, we could formulate the hyperparameter optimization problem as a stationary Lipschitz bandit after sufficient exploration in the long run. And in the very beginning we can safely believe there is also only finite number of change points. This fact firmly authenticates our problem setting and assumptions.

## A.3  Simulations for Algorithm 1

We also conduct empirical studies to evaluate our proposed Zooming TS algorithm with Restarts (Algorithm 1) in practice. Here we generate the dataset under the *switching* environment, and abruptly change the underlying mean function for several times within the time horizon $T$. The methods used for comparison as well as the simulation setting are elaborated as follows:

**Methods.** We compare our Algorithm 1 (we call it Zooming TS-R for abbreviation) with two contenders: (1) Zooming algorithm (Kleinberg et al., 2019): this algorithm is designed for the static Lipschitz bandit, and would fail in theory under the *switching* environment; (2) Oracle: we assume this algorithm knows the exact time for all switching points, and would renew the Zooming algorithm when reaching a new stationary environment. Although this algorithm could naturally perform well, but it is infeasible in reality. Therefore,

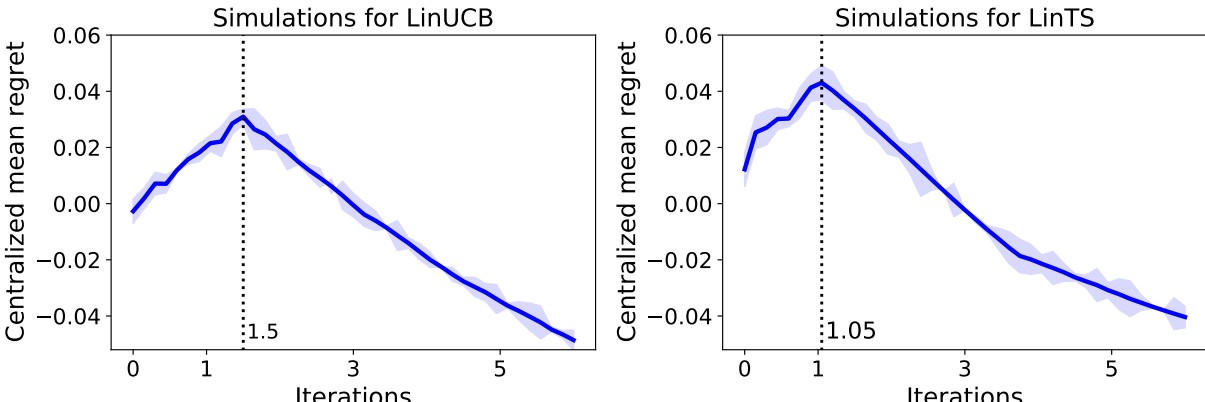

Figure 3: Average cumulative regret and its standard deviation of group mean reward for different hyperparameter values across all groups.

we would just use Oracle as a skyline here, and a direct comparison between Oracle and our Algorithm 1 is inappropriate.

**Settings.** Assume the set of arm is $[0,1]$. The unknown mean function $f_t(x)$ is chosen from two classes of reward functions with different smoothness around their maximum: (1) $\{0.9 - 0.9|x - a|, x \in [0,1] : a = 0.05, 0.25, 0.45, 0.70, 0.95\}$ (triangle function); (2) $\left\{\frac{2}{3\pi} \sin\left(\frac{3\pi}{2}(x - a + \frac{1}{3})\right), x \in [0,1] : a = 0.05, 0.25, 0.45, 0.70, 0.95\right\}$. We set $T = 90,000$ and $c(T) = 3$, and choose the location of changing points at random in the very beginning. The random noise is generated according to $N(0, 0.1)$. The value of epoch size $H$ is set as suggested by our theory $H = 10\lceil(T/c(T))^{3/4}\rceil$. For each class of reward functions, we run the simulations for 20 times and report the average cumulative regret as well as the standard deviation for each contender in Figure 4. (The change points are fixed for each repetition to make the average value meaningful.)

Figure 4 shows the performance comparisons of three different methods under the *switching* environment measured by the average cumulative regret. We can see that Oracle is undoubtedly the best since it knows the exact times for all change points and hence restart our Zooming TS algorithm accordingly. The traditional Zooming algorithm ranks the last w.r.t both mean and standard deviation since it doesn't take the non-stationarity issue into account at all, and would definitely fail when the environment changes. This fact also coincides with our expectation precisely. Our proposed algorithm has an obvious advantage over the traditional Zooming algorithm when the change points exist, and we can see that our algorithm could adapt to the environment change quickly and smoothly.

## A.4 Additional Details and Results for Section 5

### A.4.1 Baselines with A Large Candidate Set

To further make a fair comparison and validate the high superiority of our proposed CDT framework over the existing OP, TL (or Syndicated) which relies on a user-defined hyperparameter candidate set, we explore whether CDT will consistently outperform if baselines are running with a large tuning set. Here we replace the original tuning set $C_1 = \{0.1, 1, 2, 3, 4, 5\}$ with a finer set $C_2 = \{0.1, 0.25, 0.5, 0.75, 1, 1.5, 2, 2.5, 3, 3.5, 4, 4.5, 5\}$. And the new results are shown in the following Table 4 (original results in Section 5 are in gray).

Therefore, we can observe that the performance overall becomes worse under $C_2$ compared with the original $C_1$. In other words, adding lots of elements to the tuning set will not help improve the performance of existing algorithms. We believe this is because the theoretical regret bound of TL (Syndicated) also depends on the number of candidates $k$ in terms of $\sqrt{k}$ (Ding et al., 2022b). There is no theoretical guarantee for OP. After introducing so many redundant values in the candidate set, the TL (Syndicated) and OP algorithms would get disturbed and waste lots of concentration on those unnecessary candidates.

| Candidate Set | | C1 | | C2 | |
|---|---|---|---|---|---|
| Algorithm | Setting | TL/Syndicated | OP | TL/Syndicated | OP |
| LinUCB | Simulations | 343.14 | 383.62 | 356.23 | 389.91 |
| | Movielens | 346.16 | 390.10 | 359.10 | 408.67 |
| LinTS | Simulations | 828.41 | 869.30 | 874.34 | 925.29 |
| | Movielens | 519.09 | 666.35 | 516.62 | 667.77 |
| UCB-GLM | Simulations | 271.45 | 350.85 | 298.68 | 367.97 |
| | Movielens | 381.00 | 397.58 | 406.29 | 412.62 |
| GLM-TSL | Simulations | 433.27 | 445.43 | 448.21 | 458.71 |
| | Movielens | 446.74 | 678.91 | 458.23 | 718.46 |
| Laplace-TS | Simulations | 510.03 | 568.81 | 530.29 | 567.10 |
| | Movielens | 949.51 | 1063.92 | 958.10 | 1009.23 |
| GLOC | Simulations | 406.28 | 417.30 | 414.82 | 427.05 |
| | Movielens | 571.36 | 513.90 | 568.91 | 520.72 |
| SGD-TS | Simulations | 448.29 | 551.63 | 458.09 | 557.04 |
| | Movielens | 1016.72 | 1084.13 | 1038.94 | 1073.91 |

Table 4: Cumulative regrets of baselines under different hyperparameter tuning sets.

In conclusion, we believe the existing algorithms relying on user-tuned candidate sets would perform well if the size of the candidate set is reasonable and the candidate set contains some value very close to the optimal hyperparameter value. However, in practice, finding the unknown optimal hyperparameter value is a black-box problem, and it's impossible to construct a candidate set satisfying the above requirements at the beginning. If we discretize the interval finely, then the large size of the candidate set would hurt the performance as well. On the other hand, our proposed CDT could adaptively "zoom in" on the regions containing this optimal hyperparameter value automatically, without the need of pre-specifying a "good" set of hyperparameters. And CDT could always yield robust results according to the extensive experiments we did in Section 5.

On the other hand, these results also imply an interesting fact. Note it is doable to first discretize the continuous space and then implement an algorithm with discrete candidate sets, such as Syndicated (Ding et al., 2022b). However, we observe that finely discretizing the hyperparameter space will significantly hurt the practical performance and hence is wasteful and inefficient. Intuitively, it is inefficient to place lots of "probes" in other regions that do not contain the optimal point, and we should place probes in more promising regions via adaptive discretization methodology. In theory, the uniform discretization idea will lead to regret bound of order $T^{\frac{d+1}{d+2}}$ with covering dimension $d$ and the zooming idea will incur $T^{\frac{d_z+1}{d_z+2}}$ regret with zooming dimension $d_z$, and we know $d_z \le d$ and $d_z$ could be significantly smaller than $d$ under various cases. Therefore, we believe the same phenomena will occur in the non-stationary Lipschitz bandits and also our hyperparameter tuning framework as well.

### A.4.2  Ablation Study on the Choice of $T_1$ and $T_2$

For $T_1$, we set it to $T^{2/(p+3)}$ where $p$ stands for the number of hyperparameters according to Theorem 4.2. Specifically, for LinUCB, LinTS, UCB-GLM, GLM-TSL and Laplace-TS, we choose it to be 118. For GLOC and SGD-TS, we set it as 45. Here we also rerun our experiments in Section 5 with $T_1 = 0$ (no warm-up) since we believe a long warm-up period will abandon lots of useful information, and then we report the results after this change:

We can observe that the results are almost identical from Table 5. For $T_2$, Theorem 4.2 suggests that $T_2 = O\left(T^{(p+2)/(p+3)}\right)$. In our original experiments, we choose $T_2 = 3T^{(p+2)/(p+3)}$. To take an ablation study on $T_2$ we take $T_2 = kT^{(p+2)/(p+3)}$ for $k = 1, 2, 3$ in each experiment, and to see whether our CDT framework is robust to the choice of $k$.

| Algorithm | Setting | $T_1 = 0$ | $T_1 = T^{2/(p+3)}$ |
|---|---|---|---|
| LinUCB | Simulation | 298.28 | 303.14 |
|  | Movielens | 313.29 | 307.19 |
| LinTS | Simulation | 677.03 | 669.45 |
|  | Movielens | 343.18 | 340.85 |
| UCB-GLM | Simulation | 299.74 | 300.54 |
|  | Movielens | 314.41 | 311.72 |
| GLM-TSL | Simulation | 339.49 | 333.07 |
|  | Movielens | 428.82 | 432.47 |
| Laplace-TS | Simulation | 520.29 | 520.35 |
|  | Movielens | 903.16 | 900.10 |
| GLOC | Simulation | 414.70 | 418.05 |
|  | Movielens | 455.39 | 461.78 |
| SGD-TS | Simulation | 430.05 | 425.98 |
|  | Movielens | 843.91 | 838.06 |

Table 5: Ablation study on the role of $T_1$ in our CDT framework.

| Algorithm | Setting | $k = 1$ | $k = 2$ | $k = 3$ |
|---|---|---|---|---|
| LinUCB | Simulation | 328.28 | 300.62 | 298.28 |
|  | Movielens | 310.06 | 303.10 | 313.29 |
| LinTS | Simulation | 717.77 | 670.90 | 677.03 |
|  | Movielens | 360.12 | 352.19 | 343.18 |
| UCB-GLM | Simulation | 314.01 | 316.95 | 299.74 |
|  | Movielens | 347.92 | 325.58 | 314.41 |
| GLM-TSL | Simulation | 320.21 | 331.43 | 339.49 |
|  | Movielens | 439.98 | 428.91 | 428.82 |
| Laplace-TS | Simulation | 565.15 | 540.61 | 520.29 |
|  | Movielens | 948.10 | 891.91 | 903.16 |
| GLOC | Simulation | 417.05 | 414.70 | 415.05 |
|  | Movielens | 441.85 | 455.39 | 462.24 |
| SGD-TS | Simulation | 450.14 | 430.05 | 414.57 |
|  | Movielens | 852.98 | 843.91 | 830.35 |

Table 6: Ablation study on the role of $T_2$ in our CDT framework.

According to Table 6, we can observe that overall $k = 2$ and $k = 3$ perform better than $k = 1$. We believe it is because, in the long run, the optimal hyperparameter would tend to be stable, and hence some restarts are unnecessary and inefficient. Note by choosing $k = 1$ our proposed CDT still outperforms the existing TL and OP tuning algorithms overall. For $k = 2$ and $k = 3$, we can observe that their performances are comparable, which implies that the choice of $k$ is quite robust in practice. We believe it is due to the fact that our proposed Zooming TS algorithm could always adaptively approximate the optimal point. Although it is unknown which one is better in practice under different cases, our comprehensive simulations show that choosing either one in practice will work well and outperform all the existing methods. In conclusion, these results suggest that we have a universal way to set the values of $T_1$ and $T_2$ according to the theoretical bounds, and we do not need to tune them for each particular dataset. In other words, the performance of our CDT tuning framework is robust to the choice of $T_1, T_2$ under different scenarios.

## B    Supportive Remarks

*Remark* B.1. (Justifications on assumptions) We further explain the motivations of the Lipschitzness and piecewise stationarity assumptions of the expected reward function for hyperparameter tuning of bandit algorithms.

For Lipschitzness, we get the motivation of our formulation shown in Eqn. 3 and Eqn. 4 from the hyperparameter tuning work on the offline machine learning algorithms. Specifically, Bayesian optimization is widely considered as the state-of-the-art and most popular hyperparameter tuning method, which assumes that the underlying function is sampled from a Gaussian process in the given space. By selecting a value $x$ in the space and obtaining the corresponding reward, Bayesian optimization could update its estimation of the underlying function, especially in the neighbor of $x$ sequentially. And it also relies on a user-defined kernel function, whose selection is also purely empirical and lacks theoretical support. In our work, we use a similar idea as Bayesian optimization: close hyperparameters tend to yield similar values with other conditions fixed. And this natural extension motivates the Lipschitz assumption made in our paper. Therefore, it is fair to make a similar and analogous assumption (close hyperparameters yield similar results given other conditions fixed) for the hyperparameter tuning of bandit algorithms in our work. We validate this assumption using a suite of simulations in Appendix A.

For the piecewise stationarity, as we mention in Section 3, it is inappropriate to assume the strict stationarity of the bandit algorithm performance under the same hyperparameter value setting across time $T$. As an example, for most UCB and TS-based bandit algorithms (e.g. LinUCB, LinTS, UCB-GLM, GLM-UCB, GLM-TSL, etc.), the exploration degree of an arm is a multiplier of the exploration rate and the uncertainty of an arm. In the beginning, a moderate value of the exploration rate may lead to a large exploration degree for the arm since the uncertainty is large. On the contrary, in the long run, a moderate value of exploration rate will lead to a minor exploration degree for the arm since its value has been well estimated with small uncertainty. Therefore, a fixed hyperparameter setting may suggest different results across different stages of time, and hence it is unreasonable to expect the strong stationarity of the hyperparameter tuning for bandit algorithms at all time steps. On the other hand, it would be very inefficient to assume a completely non-stationary environment as in Ding et al. (2022b) which uses EXP3. In very close time steps, we could anticipate that the same hyperparameter setting would yield a very similar result in expectation since the uncertainty of any arm would be close. And using a non-stationary environment will totally waste this information and hence is inefficient. Therefore, it is very well motivated to use a partial non-stationarity assumption that lies in the middle ground between the above two extremes. Note our proposed tuning method yields very promising results in extensive experiments under our formulations. And the stationary environment can be regarded as a special case of our switching environment setting where the functions in between all change points are the same.

Finally, we will explain why it is excessively difficult to present theoretical validation regarding these assumptions in our paper. As we mentioned, our formulation is motivated by Bayesian optimization, arguably the most popular method for hyperparameter tuning for offline machine learning algorithms. And we use a similar idea: similar hyperparameters tend to yield similar values while other conditions are fixed. However, people could hardly provide any theory backing for the analogous assumption of Bayesian optimization for any offline machine learning algorithms (e.g. regression, classification), and hyperparameter tuning is widely considered as a black-box problem for offline machine learning algorithms. Not to mention that the theoretical analysis of hyperparameter tuning for any bandit algorithm is much more challenging than that of offline machine learning algorithms since historical observations along with hyperparameter values will affect the online selection simultaneously for the bandit algorithms, and we can use different hyperparameters in different rounds for bandit algorithms. Conclusively, our formulation is natural and well-motivated.

## C    Detailed Proof on the Zooming Dimension

In the beginning, we would reload some notations for simplicity. Here we could omit the time subscript (or superscript) $t$ since the following result could be identically proved for each round $t$. Assume the Lipschitz function $f$ is defined on $\mathbb{R}^{p_c}$, and $v^* \coloneqq \arg\max_{v \in A} f(v)$ denotes the maximal point (w.l.o.g. assume it's

unique), and $\Delta(v) = f(v^*) - f(v)$ is the "badness" of the arm $v$. We then naturally denote $A_r$ as the $r$-optimal region at the scale $r \in (0,1]$, i.e. $A_r = \{v \in A : r/2 < \Delta(v) \leq r\}$. The $r$-zooming number could be denoted as $N_z(r)$. And the zooming dimension could be naturally denoted as $p_z$. Note that by the Assouad's embedding theorem, any compact doubling metric space $(A, \mathrm{Dist}(\cdot,\cdot))$ can be embedded into the Euclidean space with some type of metric. Therefore, for all compact doubling metric spaces with cover dimension $p_c$, it is sufficient to study on the metric space $([0,1]^{p_c}, \|\cdot\|^l)$ for some $l \in (0, +\infty]$ instead.

We will rigorously prove the following two facts regarding the $r$-zooming number $N_z(r)$ of $(A, f)$ for arbitrary compact set $A \subseteq \mathbb{R}^{p_c}$ and Lipschitz function $f(\cdot)$ defined on $A$:

- $0 \leq p_z \leq p_c$.

- The zooming dimension could be much smaller than $p_c$ under some mild conditions. For example, if the payoff function $f$ is greater than $\|v^* - v\|^\beta$ in scale in a (non-trivial) neighborhood of $v^*$ for some $\beta \geq 1$, i.e. $f(v^*) - f(v) \geq C(\|v^* - v\|^\beta)$ as $\|v^* - v\| \leq r$ for some $C > 0$ and $r = \Theta(1)$, then it holds that $p_z \leq (1 - 1/\beta)p_c$. Note $\beta = 2$ when we have $f(\cdot)$ is $C^2$-smooth and strongly concave in a neighborhood of $v^*$, which subsequently implies that $p_z \leq p_c/2$.

*Proof.* Due to the compactness of $A$, it suffices to prove the results when $A = [0,1]^{p_c}$. By the definition of the zooming dimension $p_z$, it naturally holds that $p_z \geq 0$. On the other side, since the space $A$ is a closed and bounded set in $\mathbb{R}^{p_c}$, we assume the radius of $A$ is no more than $S$, which consequently implies that the $r/16$-covering number of $A$ is at most the order of

$$\left(\frac{S}{\frac{r}{16}}\right)^{p_c} = (16S)^{p_c} \cdot r^{-p_c}.$$

Since we know $A_r \subseteq A$, it holds that $p_z \leq p$. Secondly, if the payoff function $f$ is locally greater than $\|v^* - v\|^\beta$ in scale for some $\beta \geq 1$, i.e. $f(v^*) - f(v) \geq C(\|v^* - v\|^\beta)$, then there exists $C \in \mathbb{R}$ and $\delta > 0$ such that as long as $C \|v - v^*\|^\beta \leq \delta$ we have $f(v^*) - f(v) \geq C \|v - v^*\|^\beta$. Therefore, for $0 < r < \delta$, it holds that,

$$\{v : r \geq f(v^*) - f(v) > r/2\} \subseteq \{v : C \|v - v^*\|^\beta \leq r\} = \left\{v : \|v - v^*\| \leq \left(\frac{r}{C}\right)^{\frac{1}{\beta}}\right\}$$

It holds that the $r$-covering number of the Euclidean ball with center $v^*$ and radius $(r/c)^{(1/\beta)}$ is of the order of

$$\left(\frac{\left(\frac{r}{C}\right)^{\frac{1}{\beta}}}{\frac{r}{16}}\right)^{p_c} = \left(\frac{16}{C^{\frac{1}{\beta}}}\right)^{p_c} \cdot r^{-(1 - \frac{1}{\beta})p_c}$$

which explicitly implies that $p_z \leq (1 - 1/\beta)p_c$. $\qquad\square$

## D   Intuition of our Thompson Sampling update

Intuitively, we consider a Gaussian likelihood function and Gaussian conjugate prior to design our Thompson Sampling version of zooming algorithm, and here we would ignore the clipping step for explanation. Suppose the likelihood of reward $\tilde{y}_t$ at time $t$, given the mean of reward $I(v_t)$ for our pulled arm $v_t$, follows a Gaussian distribution $N(I(v_t), s_0^2)$. Then, if the prior of $I(v_t)$ at time $t$ is given by $N(\hat{f}_t(v_t), s_0^2/n_t(v_t))$, we could easily compute the posterior distribution at time $t + 1$,

$$\mathrm{Pr}(I(v_t)|\tilde{y}_t) \propto \mathrm{Pr}(\tilde{y}_t|I(v_t))\mathrm{Pr}(I(v_t)),$$

as $N(\hat{f}_{t+1}(v_t), s_0^2/n_{t+1}(v_t))$. We can see this result coincides with our design in Algorithm 1 and its proof is as follows:

Figure 4: Cumulative regret plots of Zooming TS-R, Zooming and Oracle algorithms under the *switching* environment.

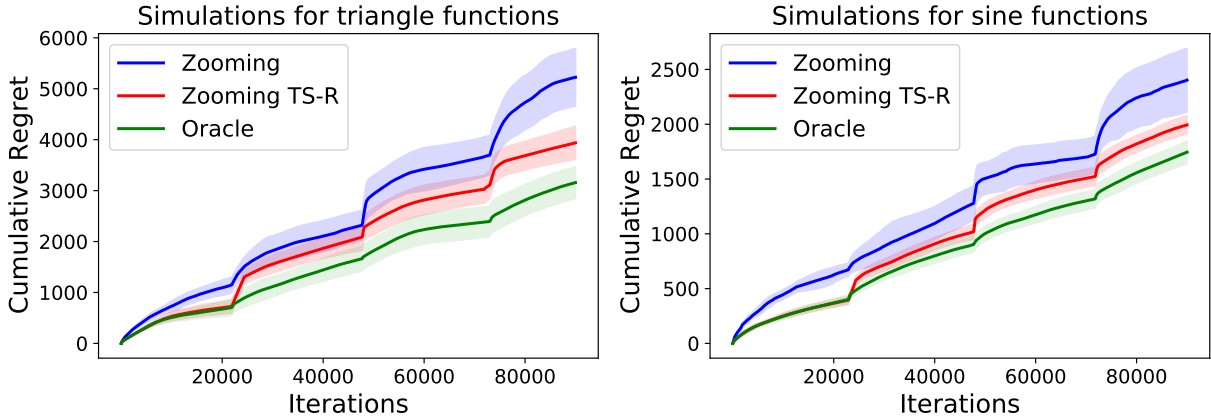

*Proof.*

$$\Pr(I(v_t)|\tilde{y}_t) \propto \Pr(\tilde{y}_t|I(v_t))\Pr(I(v_t))$$

$$\propto \exp\left\{-\frac{1}{2s_0^2}[(I(v_t) - \tilde{y}_t)^2 + n_t(v_t)(I(v_t) - f_t(v_t))^2]\right\}$$

$$\propto \exp\left\{-\frac{1}{2s_0^2}[(n_t(v_t) + 1)I(v_t)^2 - 2(\tilde{y}_t + n_t(v_t)f_t(v_t))I(v_t)]\right\}$$

$$\propto \exp\left\{-\frac{n_{t+1}(v_t)}{2s_0^2}\left[I(v_t)^2 - 2\frac{(\tilde{y}_t + n_t(v_t)f_t(v_t))}{n_{t+1}(v_t)}I(v_t)\right]\right\}$$

$$\propto \exp\left\{-\frac{n_{t+1}(v_t)}{2s_0^2}(I(v_t) - f_{t+1}(v_t))^2\right\}$$

Therefore, the posterior distribution of $I(v_t)$ at time $t + 1$ is $N(f_{t+1}(v_t), s_0^2\frac{1}{n_{t+1}(v_t)})$. □

This gives us an intuitive explanation why our Zooming TS algorithm works well when we ignore the clipped distribution step. And we have stated that this clipping step is inevitable in Lipschitz bandit setting in our main paper since (1) we'd like to avoid underestimation of good active arms, i.e. avoid the case when their posterior samples are too small. (2) We could at most adaptively zoom in the regions which contains $v^*$ instead of exactly detecting $v^*$, and this inevitable loss could be mitigated by setting a lower bound for TS posterior samples. Note that although the intuition of our Zooming TS algorithm comes from the case where contextual bandit rewards follow a Gaussian distribution, we also prove that our algorithm can achieve a decent regret bound under the *switching* environment and the optimal instance-dependent regret bound under the stationary Lipschitz bandit setting.

# E    Proof of Theorem 4.1

## E.1    Stationary Environment Case

To prove Theorem 4.1, we will first focus on the stationary case, where $f_t := f, \forall t \in [T]$. When the environment is stationary, we could omit the subscript (or superscript) $t$ in some notations as in Section C for simplicity: Assume the Lipschitz function is $f$, and $v^* := \arg\max_{v \in A} f(v)$ denotes the maximal point (w.l.o.g. assume it's unique), and $\Delta(v) = f(v^*) - f(v)$ is the "badness" of the arm $v$. We then naturally denote $A_r$ as the $r$-optimal region at the scale $r \in (0, 1]$, i.e. $A_r = \{v \in A : r/2 < \Delta(v) \le r\}$. The $r$-zooming number could be denoted as $N_z(r)$. And the zooming dimension could be naturally denoted as $p_z$. Note we could omit the subscript (or superscript) $t$ for the notations just mentioned above since all these values would be fixed through all rounds under the stationary environment.

### E.1.1 Useful Lemmas and Corollaries

Recall that $\hat{f}_t(v)$ is the average observed reward for arm $v \in A$ by time $t$. And we call all the observations (pulled arms and observed rewards) over $T$ total rounds as a process.

**Definition E.1.** We call it a clean process, if for each time $t \in [T]$ and each strategy $v \in A$ that has been played at least once at any time $t$, we have $|\hat{f}_t(v) - f(v)| \le r_t(v)$.

**Lemma E.2.** *The probability that, a process is clean, is at least $1 - 1/T$.*

*Proof.* Fix some arm $v$. Recall that each time an algorithm plays arm $v$, the reward is sampled IID from some distribution $\mathbb{P}_v$. Define random variables $U_{v,s}$ for $1 \le s \le T$ as follows: for $s \le n_T(v)$, $U_{v,s}$ is the reward from the $s$-th time arm $v$ is played, and for $s > n_T(v)$ it is an independent sample from $\mathbb{P}_v$. For each $k \le T$ we can apply Chernoff bounds to $\{U_{v,s} : 1 \le s \le k\}$ and obtain that:

$$\Pr\left( \left| \frac{1}{k} \sum_{s=1}^{k} U_{v,s} - f(v) \right| \ge \sqrt{\frac{13\tau_0^2 \ln T}{2k}} \right) \le 2 \cdot \exp\left( -\frac{k}{2\tau_0^2} \frac{13\tau_0^2 \ln T}{2k} \right)$$

$$= 2 \exp\left( \frac{13}{4} \ln T \right) = 2T^{-3.25} \le T^{-3}, \tag{7}$$

since we can trivially assume that $T \ge 16$. Let $N$ be the number of arms activated all over rounds $T$; note that $N \le T$. Define $X$-valued random variables $\{x_i\}_{i=1}^{T}$ as follows: $x_j$ is the $\min(j, N)$-th arm activated by time $T$. For any $x \in A$ and $j \le T$, the event $\{x = x_j\}$ is independent of the random variables $\{U_{x,s}\}$: the former event depends only on payoffs observed before $x$ is activated, while the latter set of random variables has no dependence on payoffs of arms other than $x$. Therefore, Eqn. equation 7 is still valid if we replace the probability on the left side with conditional probability, conditioned on the event $\{x = x_j\}$. Taking the union bound over all $k \le T$, it follows that:

$$\Pr(\forall t \le T, |f(v) - \hat{f}_t(v)| \le r_t(v) \,|\, x_j = v) \ge 1 - T^{-2}, \quad \forall v \in A, j \in [T],$$

Integrating over all arms $v$ we get

$$\Pr(\forall t \le T, |f(x_j) - \hat{f}_t(x_j)| \le r_t(x_j)) \ge 1 - T^{-2}, \quad \forall j \in [T].$$

Finally, we take the union bound over all $j \le T$, and it holds that,

$$\Pr(\forall t \le T, j \le T, |f(x_j) - \hat{f}_t(x_j)| \le r_t(x_j)) \ge 1 - T^{-1},$$

and this obviously implies the result. □

**Lemma E.3.** *If it is a clean process, then $B(v, r_t(v))$ could never be eliminated from Algorithm 1 for any $t \in [T]$ and arm $v$ that is active at round $t$, given that $v^* \in B(v, r_t(v))$.*

*Proof.* Recall that from Algorithm 1, at round $t$ the ball $B(u, r_t(u))$ would be permanently removed if we have for some active arm $v$ s.t.
$$\hat{f}_t(v) - r_t(v) > \hat{f}_t(u) + 2r_t(u).$$
If we have that $v^* = \arg\max_{x \in A} f(x) \in B(u, r_t(u))$, then it holds that

$$\hat{f}_t(u) + 2r_t(u) \ge f(u) + r_t(u) \ge f(u) + \mathrm{Dist}(u, v^*) \ge f(v^*),$$

where the first inequality is due to the clean process and the last one comes from the fact that $f$ is a Lipschitz function. On the other hand, we have that for any active arm $v$,

$$f(v) \ge \hat{f}_t(v) - r_t(v), \quad f(v^*) \ge f(v).$$

Therefore, it holds that

$$\hat{f}_t(v) - r_t(v) \le \hat{f}_t(u) + 2r_t(u).$$

And this inequality concludes our proof. □

**Lemma E.4.** *If it is a clean process, then for any time $t$ and any active strategy $v$ that has been played at least once before time $t$ we have $\Delta(v) \leq 5\mathbb{E}[r_t(v)]$. Furthermore, it holds that $\mathbb{E}(n_t(v)) \leq O(\ln(T)/\Delta(v)^2)$.*

*Proof.* Let $S_t$ be the set of all arms that are active at time $t$. Suppose an arm $v_t$ is played at time $t$ and was previously played at least twice before time $t$. Firstly, We would claim that

$$f(v^*) \leq I_t(v_t) \leq f(v_t) + 3r_t(v_t)$$

holds uniformly for all $t$ with probability at least $1 - \delta$, which directly implies that $\Delta(v_t) \leq 3r_t(v_t)$ with high probability uniformly. First we show that $I_t(v_t) \geq f(v^*)$. Indeed, recall that all arms are covered at time $t$, so there exists an active arm $v_t^*$ that covers $v^*$, meaning that $v^*$ is contained in the confidence ball of $v_t^*$. And based on Lemma E.3 the confidence ball containing $v^*$ could never be eliminated at round $t$ when it's a clean process. Recall $Z_{t,v}$ is the i.i.d. standard normal random variable used for any arm $v$ in round $t$ (Eqn. equation 6). Since arm $v_t$ was chosen over $v_t^*$, we have $I_t(v_t) \geq I_t(v_t^*)$. Since this is a clean process, it follows that

$$I_t(v_t^*) = \hat{f}_t(v_t^*) + s_0\sqrt{\frac{1}{n_t(v_t^*)}} Z_{t,v_t^*} \geq f(v_t^*) + s_0\sqrt{\frac{1}{n_t(v_t^*)}} Z_{t,v_t^*} - r_t(v_t^*) \tag{8}$$

Furthermore, according to the Lipschitz property we have

$$f(v_t^*) \geq f(v^*) - \text{Dist}(v_t^*, v^*) \geq f(v^*) - r_t(v_t^*). \tag{9}$$

Combine Eqn. equation 8 and equation 9, we have

$$I_t(v_t) \geq I_t(v_t^*) \geq f(v^*) + s_0\sqrt{\frac{1}{n_t(v_t^*)}} Z_{t,v_t^*} - 2r_t(v_t^*)$$

$$= f(v^*) + \sqrt{\frac{52\pi\tau_0^2 \ln(T)}{n_t(v_t^*)}} \left( Z_{t,v_t^*} - \frac{1}{\sqrt{2\pi}} \right) \geq f(v^*), \tag{10}$$

where we get the last inequality since we truncate the random variable $Z_{t,v_t^*}$ by the lower bound $1/\sqrt{2\pi}$ according to the definition. On the other hand, we have

$$I_t(v_t) \leq f(v_t) + r_t(v_t) + s_0\sqrt{\frac{1}{n_t(v_t)}} Z_{t,v_t} = f(v_t) + \left( 1 + 2\sqrt{2\pi} Z_{t,v_t} \right) r_t(v_t) \tag{11}$$

Therefore, by combining Eqn. equation 10 and equation 11 we have that

$$\Delta(v_t) \leq \left( 1 + 2\sqrt{2\pi} Z_{t,v_t} \right) r_t(v_t). \tag{12}$$

And we know that $Z_{t,:}$ is defined as $Z_{t,:} = \max\{1/\sqrt{2\pi}, \tilde{Z}_{t,:}\}$ where $\tilde{Z}_{t,:}$ is IID drawn from standard normal distribution. In other words, $Z_{t,v_t}$ follows a clipped normal distribution with the following PDF:

$$f(x) = \begin{cases} \phi(x) + (1 - \Phi(x))\delta\left(x - \frac{1}{\sqrt{2\pi}}\right), & x \geq \frac{1}{\sqrt{2\pi}}; \\ 0, & x < \frac{1}{\sqrt{2\pi}}; \end{cases}$$

Here $\phi(\cdot)$ and $\Phi(\cdot)$ denote the PDF and CDF of standard normal distribution. And we have

$$\mathbb{E}(Z_{t,v_t}) \leq \frac{1}{\sqrt{2\pi}} + \int_{\frac{1}{\sqrt{2\pi}}}^{+\infty} x\phi(x)dx \leq \frac{1}{\sqrt{2\pi}} + \frac{1}{\sqrt{2\pi}} e^{-\frac{1}{4\pi}} \leq \sqrt{\frac{2}{\pi}}$$

By taking expectation on Eqn. equation 12, we have $\Delta(v_t) \leq 5\mathbb{E}(r_t(v_t))$. Next, we would show that $\mathbb{E}(n_t(v_t)) \leq O(\ln(T))/\Delta(v_t)^2$. Based on Eqn. equation 11 and the definition of $r_t(\cdot)$, we could deduce that

$$\sqrt{n_t(v_t)} \leq \sqrt{\frac{13}{2}\tau_0^2 \ln(T)}(1 + 2\sqrt{2\pi} Z_{t,v_t})\frac{1}{\Delta(v_t)},$$

which thus implies that

$$n_t(v_t) \le \frac{13}{2}\tau_0^2 \ln{(T)}(1 + 2\sqrt{2\pi}Z_{t,v_t})^2 \frac{1}{\Delta(v_t)^2} = O(\ln{(T)})(1 + 2\sqrt{2\pi}Z_{t,v_t})^2 \frac{1}{\Delta(v_t)^2}. \tag{13}$$

By simple calculation, we could show that

$$\mathbb{E}(Z_{t,v_t}^2) \le \frac{1}{2\pi} + \int_{\frac{1}{\sqrt{2\pi}}}^{+\infty} x^2 \phi(x)dx \le \frac{1}{\pi} + \frac{1}{2} \le 1$$

$$\Rightarrow \quad \mathbb{E}\left[(1 + 2\sqrt{2\pi}Z_{t,v_t})^2\right] \le 1 + 4\sqrt{2\pi}\sqrt{\frac{2}{\pi}} + 8\pi < +\infty.$$

After revisiting Eqn. equation 13, we can show that $\mathbb{E}(n_t(v_t)) \le O(\ln{(T)})/\Delta(v_t)^2$. Now suppose arm $v$ is only played once at time $t$, then $r_t(v) > 1$ and thus the lemma naturally holds. Otherwise, let $s$ be the last time arm $v$ has been played according to the selection rule, where we have $r_t(v) = r_s(v)$, and then based on Eqn. equation 11 it holds that

$$I_t(v) \le f(v) + \left(1 + 2\sqrt{2\pi}Z_{s,v}\right)r_t(v).$$

And then we could show that $\Delta(v) \le 5\mathbb{E}(r_t(v))$. By using an identical argument as before, we could show that $\mathbb{E}(n_t(v)) \le O(\ln{(T)})/\Delta(v)^2$. $\qquad\square$

**Lemma E.5.** *Let $X_1, \ldots, X_n$ be independent $\sigma^2$-sub-Gaussian random variables. Then for every $t > 0$,*

$$P\left(\max_{1,\le,n} X_i \ge \sqrt{2\sigma^2(\ln{(T)} + t)}\right) \le e^{-t}.$$

*Proof.* Let $u = \sqrt{2\sigma^2(\ln{(n)} + t)}$, we have

$$P\left(\max_{1,\le,n} X_i \ge u\right) = P(\exists i, X_i \ge u) \le \sum_{i=1}^n P(X_i \ge u) \le ne^{-\frac{u^2}{2\sigma^2}} = e^{-t}.$$

$\qquad\square$

### E.1.2   Proof of Theorem 4.1 under stationary environment

*Proof.* By Lemma E.2 we know that it is a clean process with probability at least $1 - \frac{1}{T}$. In other words, denote the event $\Omega \coloneqq \{\text{clean process}\}$, and then we have that $P(\Omega) \ge 1 - \frac{1}{T}$. And according to Lemma E.3 we're aware that the active confidence balls containing the best arm can't be removed in a clean process. Remember that we use $S_T$ as the set of all arms that are active in the end, and denote

$$B_{i,T} = \left\{v \in S_T : 2^i \le \frac{1}{\Delta(v)} < 2^{i+1}\right\}, \quad \text{where} \quad S_T = \bigcup_{i=0}^{+\infty} B_{i,T},$$

where $i \ge 0$. Then, under the event $\Omega$, by using Corollary E.4 we have $\mathbb{E}(n_T(v)|\Omega) \le O(\ln T)/\Delta(v)^2$, and hence it holds that

$$\sum_{v \in B_{i,T}} \Delta(v)\mathbb{E}(n_T(v)|\Omega) \le O(\ln T)\sum_{v \in B_{i,t}} \frac{1}{\Delta(v)} \le O(\ln T) \cdot 2^i |B_{i,t}|$$

Denote $r_i = 2^{-i}$, we have

$$\sum_{v \in B_{i,T}} \Delta(v)\mathbb{E}(n_T(v)|\Omega) \le O(\ln T) \cdot \frac{1}{r_i}|B_{i,t}|$$

Next, we would show that for any active arms $u, v$ we have

$$\mathrm{Dist}(u,v) > \frac{1}{4\sqrt{2\pi \ln(T)}} \min\{\Delta(u), \Delta(v)\} \tag{14}$$

with probability at least $1 - \frac{1}{T}$. W.l.o.g assume $u$ has been activated before $v$. Let $s$ be the time when $v$ has been activated. Then by the philosophy of our algorithm we have that $\mathrm{Dist}(u,v) > r_s(v)$. Then according to Eqn. equation 12 in the proof Lemma E.4, it holds that $r_s(v) \geq \frac{1}{2\sqrt{2\pi}Z}\Delta(v)$ for some random variable $Z$ following the clipped standard normal distribution. Define the event $\Upsilon = \{Z_{t,v_t} < 2\sqrt{\ln(T)} \text{ for all } t \in [T]\}$, then based on Lemma E.5 we have $P(\Upsilon) \geq 1 - \frac{1}{T}$. Then under the event $\Upsilon$, we have $r_s(v) \geq \frac{1}{4\sqrt{2\pi \ln(T)}}\Delta(v)$, which then implies that Eqn. equation 14 holds under $\Upsilon$. Since for arbitrary $x, y \in B_{i,T}$ we have

$$\frac{r_i}{2} < \Delta(x) \leq r_i, \quad \frac{r_i}{2} < \Delta(y) \leq r_i,$$

which implies that under the event $\Upsilon$

$$\mathrm{Dist}(x,y) > \frac{1}{4\sqrt{2\pi \ln(T)}} \min\{\Delta(x), \Delta(y)\} > \frac{r_i}{8\sqrt{2\pi \ln(T)}}.$$

Therefore, $x$ and $y$ should belong to different sets of $(r_i/8\sqrt{2\pi \ln(T)})$-diameter-covering. It follows that $|B_{i,T}| \leq N_z(r_i/8\sqrt{2\pi \ln(T)}) \leq O(\ln(T)^p)cr_i^{p_z} \leq \tilde{O}(cr_i^{p_z})$. Recall $N_z(r)$ is defined as the minimal number of balls of radius no more than $r$ required to cover $A_r$. As a result, under the events $\Omega$ and $\Upsilon$, it holds that

$$\sum_{v \in B_{i,T}} \Delta(v)\mathbb{E}(n_T(v) \mid \Omega \cap \Upsilon) \leq O(\ln T) \cdot \frac{1}{r_i}N_z(r_i) \tag{15}$$

Therefore, based on Eqn. equation 15, we have

$$
\begin{aligned}
R_L(T) &= \sum_{v \in S_T} \Delta(v)\mathbb{E}(n_T(v)) \\
&= P(\Omega \cap \Upsilon)\sum_{v \in S_T}\Delta(v)\mathbb{E}(n_T(v)\mid \Omega \cap \Upsilon) + P(\Omega^c \cup \Upsilon^c)\sum_{v \in S_T}\Delta(v)\mathbb{E}(n_T(v)\mid \Omega^c \cup \Upsilon^c) \\
&\leq \sum_{v \in S_T : \Delta(v) \leq \rho}\Delta(v)\mathbb{E}(n_T(v)\mid \Omega \cap \Upsilon) + \sum_{v \in S_T : \Delta(v) > \rho}\Delta(v)\mathbb{E}(n_T(v)\mid \Omega \cap \Upsilon) + \frac{2}{T}\cdot T \\
&\leq \rho T + \sum_{i < \log_2(\frac{1}{\rho})}\frac{1}{r_i}\tilde{O}(cr_i^{-p_z}) + 2 \\
&\leq \rho T + \tilde{O}(1)\sum_{i < \log_2(\frac{1}{\rho})}\frac{1}{r_i}cr_i^{-p_z} + 2 \\
&\leq \rho T + \tilde{O}(1)\sum_{k=0}^{\lfloor \log_{1/2} 2\rho \rfloor}c2^{k(p_z+1)} + 2 \\
&\leq \rho T + \tilde{O}(1)\cdot 2\cdot 2^{\lfloor \log_{1/2} 2\rho \rfloor (p_z+1)} + 2 \\
&\leq \rho T + \tilde{O}(1)\left(\frac{1}{2\rho}\right)^{p_z+1} + 2
\end{aligned}
$$

By choosing $\rho$ in the scale of

$$\rho = \tilde{O}\left(\frac{1}{T}\right)^{\frac{1}{p_z+2}},$$

it holds that

$$R_L(T) = \tilde{O}\left(T^{\frac{p_z+1}{p_z+2}}\right).$$

### E.2  *Switching* (Non-stationary) Environment Case

Since there are $c(T)$ change points for the environment Lipschitz functions $f_t(\cdot)$, i.e.

$$\sum_{t=1}^{T-1} \mathbf{1}[\exists m \in A \,:\, f_t(m) \neq f_{t+1}(m)] = c(T).$$

Given the length of epochs as $H$, we would have $\lceil T/H \rceil$ epochs overall. And we know that among these $\lceil T/H \rceil$ different epochs, at most $c(T)$ of them contain the change points. For the rest of epochs that are free of change points, the cumulative regret could be bounded by the result we just deduced for the stationary case above. And the cumulative regret in any epoch with stationary environment could be bounded as $H^{(p_{z,*}+1)/(p_{z,*}+2)}$. Specifically, we could partition the $T$ rounds into $m = \lceil T/H \rceil$ epochs:

$$[T_1 + 1, T] = [\omega_0 = T_1 + 1, \omega_1) \cup [\omega_1, \omega_2) \cup \cdots \cup [\omega_{m-1}, \omega_m = T + 1),$$

where $\omega_{i+1} = \omega_i + H$ for $i = 0, \ldots, m-2$. Denote all the change points as $T_1 \leq \rho_1 < \cdots < \rho_{c(T)} \leq T$, and then define

$$\Omega = \{ \cup [\omega_i, \omega_{i+1}) \,:\, \rho_j \in [\omega_i, \omega_{i+1}), \exists j = 1, \ldots c; \, i = 0, \ldots, m-1 \}.$$

Then it holds that $|\Omega| \leq Hc(T)$. Therefore, it holds that

$$R_L(T) \leq \tilde{O}\left( Hc(T) + \left(\frac{T}{H} + 1\right) H^{\frac{p_{z,*}+1}{p_{z,*}+2}} \right) \leq \tilde{O}\left( Hc(T) + \frac{T}{H} \cdot H^{\frac{p_{z,*}+1}{p_{z,*}+2}} \right),$$

where the first part bound the regret of non-stationary epochs and the second part bound that of stationary ones. By taking $H = (T/c(T))^{(p_{z,*}+2)/(p_{z,*}+3)}$, it holds that

$$R_L(T) \leq \tilde{O}\left( T^{\frac{p_{z,*}+2}{p_{z,*}+3}} c(T)^{\frac{1}{p_{z,*}+3}} \right).$$

And this concludes our proof for Theorem 4.1.  $\square$

## F  Algorithm 1 with unknown $c(T)$ and $p_{z,*}$

### F.1  Introduction of Algorithm 3

When both the number of change points $c(T)$ over the total time horizon $T$ and the zooming dimension $p_{z,*}$ are unknown, we could adapt the BOB idea used in Cheung et al. (2019); Zhao et al. (2020) to choose the optimal epoch size $H$ based on the EXP3 meta algorithm. In the following, we first describe how to use the EXP3 algorithm to choose the epoch size dynamically even if $c(T)$ and $p_{z,*}$ are unknown. Then we present the regret analysis in Theorem F.1 and its proof.

Although the zooming dimension $p_{z,*}$ is unknown, it holds that $p_{z,*} \leq p_c$, and hence we could simply use the upper bound of $p_{z,*}$ (denoted as $p_u$) as $p_c$ instead (recall $p_c$ is the covering dimension). Note that the upper bound $p_{z,*}$ could be more specific when we have some prior knowledge of the reward Lipschitz function $f(\cdot)$: for example, as we mentioned in Appendix C, if the function $f(\cdot)$ is known to be $C^2-$smooth and strongly concave in a neighborhood of its maximum defined in $\mathbb{R}^{p_c}$, it holds that $p_{z,*} \leq p_c/2$ and then we could use $p_u = p_c/2$ as the upper bound. Note that we also use the BOB mechanism in the CDT framework for hyperparameter tuning in Algorithm 2, where we treat the zooming TS algorithm with Restarts as the meta algorithm to select the hyperparameter setting in the upper layer, and then use the selected configuration for the bandit algorithm in the lower layer. However, here we would use BOB mechanism differently: we firstly divide the total horizon $T$ into several epochs of the same length $H_0$ (named top epoch), where in each top epoch we would restart the Algorithm 1. And in the $i-$th top epoch the restarting length $H_i$ (named bottom epoch) of Algorithm 1 could be chosen from the set $J = \{J_i := \lceil k \rceil \,:\, k \geq 1, k = H_0/2^{i-1}, i = 1, 2, \ldots \}$, where the chosen bottom epoch size could be adaptively tuned by using EXP3 as the meta algorithm. Here we

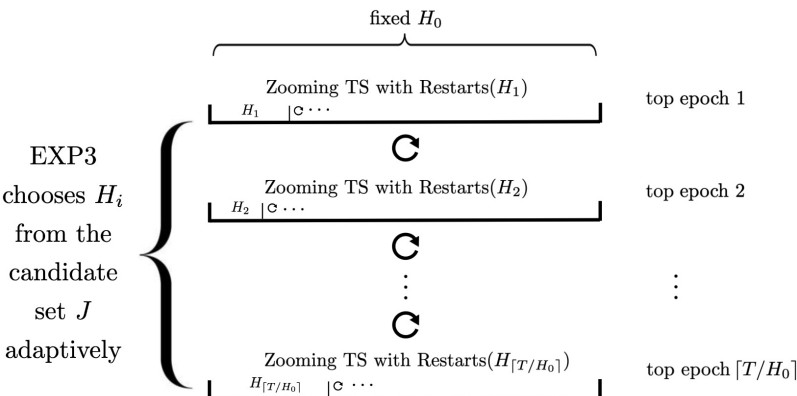

Figure 5: An illustration of Zooming TS algorithm with double restarts when $c(T)$ is agnostic.

restart the zooming TS algorithm from two perspectives, where we first restart the zooming TS algorithm with Restarts (Algorithm 1) in each top epoch of some fixed length $H_0$, and then for each top epoch the restarting length $H_i$ for Algorithm 1 would be tuned on the fly based on the previous observations (Cheung et al., 2019). Therefore, we would name this method Zooming TS algorithm with Double Restarts.

As for how to choose the bottom epoch size $H_i$ in each top epoch of length $H_0$, we implement a two-layer framework: In the upper layer, we use the adversarial MAB algorithm EXP3 to pull the candidate from $J = \{J_i\}$. And then in the lower layer we use it as the bottom epoch size for Algorithm 1. When a top epoch ends, we would update the components in EXP3 based on the rewards witnessed in this top epoch. The illustration of this double restarted strategy is depicted in Figure 5. And the detailed procedure is shown in Algorithm 3.

**Theorem F.1.** *By using the (top) epoch size as $H_0 = \lceil T^{(p_u+2)/(p_u+4)} \rceil$, the expected total regret of our Zooming TS algorithm with Double Restarts (Algorithm 3) under the switching environment over time $T$ could be bounded as*

$$R_L(T) \leq \tilde{O}\left(T^{\frac{p_u+2}{p_u+3}} \cdot \max\left\{c(T)^{\frac{1}{p_u+3}}, T^{\frac{1}{(p_u+3)(p_u+4)}}\right\}\right).$$

*Specifically, it holds that*

$$R_L(T) \leq \begin{cases} T^{\frac{p_u+2}{p_u+3}} c(T)^{\frac{1}{p_u+3}}, & c(T) \geq T^{\frac{1}{p_u+4}}, \\ T^{\frac{p_u+3}{p_u+4}}, & c(T) < T^{\frac{1}{p_u+4}}, \end{cases}$$

*where $p_u \leq p_c$ is the upper bound of $p_{z,*}$.*

Therefore, we observe that if $c(T)$ is large enough, we could obtain the same regret bound as in Theorem 4.1 given $p_{z,*}$.

### F.2   Proof of Theorem F.1

*Proof.* The proof of Theorem F.1 relies on the recent usage of the BOB framework that was firstly introduced in Cheung et al. (2019) and then widely used in various bandit-based model selection work (Ding et al., 2022a; Zhao et al., 2020). To be consistent we would use the notations in Algorithm 3 in this proof, and we would also recall these notations here for readers' convenience: for the $i$-th bottom epoch, we assume the candidate $H_{j_i}$ is pulled from the set $J$ in the beginning, where $j_i$ is the index of the pulled candidate. At round $t$, given the current bottom epoch length $H_{j_i}$ for some $i$, we pull the arm $v_t(H_{j_i}) \in A$ and then collect the stochastic reward $Y_t$. We also define $c_i(T)$ as the number of change points during each top epoch, and hence it naturally holds that $\sum_{i=1}^{\lceil T/H_0 \rceil} c_i(T) = c(T)$. Given these notations, the expected cumulative regret

could be decomposed into the following two parts:

$$R_L(T) = \mathbb{E}\left[\sum_{t=1}^{T} f_t(v_t^*) - f_t(v_t)\right] = \mathbb{E}\left[\sum_{i=1}^{\lceil T/H_0\rceil} \sum_{t=(i-1)H_0+1}^{\min\{T,iH_0\}} f_t(v_t^*) - f_t(v_t(H_{j_i}))\right]$$

$$= \underbrace{\mathbb{E}\left[\sum_{i=1}^{\lceil T/H_0\rceil} \sum_{t=(i-1)H_0+1}^{\min\{T,iH_0\}} f_t(v_t^*) - f_t(v_t(H^*))\right]}_{\text{Quantity (I)}}$$

$$+ \underbrace{\mathbb{E}\left[\sum_{i=1}^{\lceil T/H_0\rceil} \sum_{t=(i-1)H_0+1}^{\min\{T,iH_0\}} f_t(v_t(H^*)) - f_t(v_t(H_{j_i}))\right]}_{\text{Quantity (II)}}, \tag{16}$$

where $H^*$ could be any restarting period in $J$, and we expect it could approximate the optimal choice $H^{\text{opt}} = (T/c(T))^{(p_u+2)/(p_u+3)}$ in Theorem 4.1. (Here we replace $p_{z,*}$ by $p_u$ in Theorem 4.1 since the underlying $p_{z,*}$ is mostly unspecified in reality.) According to the proof of Theorem 4.1 in Appendix G, the Quantity (I) could be bounded as:

$$\mathbb{E}\left[\sum_{i=1}^{\lceil T/H_0\rceil} \sum_{t=(i-1)H_0+1}^{\min\{T,iH_0\}} f_t(v_t^*) - f_t(v_t(H^*))\right] \leq \sum_{i=1}^{\lceil T/H_0\rceil} H^* c_i(T) + \frac{H_0}{H^*}(H^*)^{\frac{p_u+2}{p_u+4}}$$

$$= H^* c(T) + T(H^*)^{-\frac{1}{p_u+2}}$$

However, it is clear that each candidate in $J$ could at most be the length of top epoch size $H_0$, which we set to be $\lceil T^{(p_u+2)/(p_u+4)}\rceil$, and hence it would be more challenging if the optimal choice $H^{\text{opt}} = (T/c(T))^{(p_u+2)/(p_u+3)}$ is larger than $H_0$. To deal with this issue, we bound the expected cumulative regret in two different cases separately:

(1) If $H^{\text{opt}} = (T/c(T))^{(p_u+2)/(p_u+3)} \leq H_0$, which is equivalent to

$$\left(\frac{T}{c(T)}\right)^{\frac{p_u+2}{p_u+3}} \leq H_0 \Leftrightarrow \left(\frac{T}{c(T)}\right)^{\frac{p_u+2}{p_u+3}} \leq T^{\frac{p_u+2}{p_u+4}} \Leftrightarrow c(T) \geq T^{\frac{1}{p_u+4}},$$

then we know that there exists some $H^+ \in J$ such that $H^+ \leq (T/c(T))^{(p_u+2)/(p_u+3)} \leq 2H^+$. By setting $H^* = H^+$, the Quantity (I) could be bounded as:

$$\text{Quantity (I)} = \tilde{O}\left(H^+ c(T) + T(H^*)^{-\frac{1}{p_u+2}}\right)$$

$$= \tilde{O}\left(H^{\text{opt}} c(T) + T(H^{\text{opt}})^{-\frac{1}{p_u+2}}\right) = \tilde{O}\left(T^{\frac{p_u+2}{p_u+3}} c(T)^{\frac{1}{p_u+3}}\right).$$

For the Quantity (II), we could bound it based on the results in Auer et al. (2002b). Specifically, from Corollary 3.2 in Auer et al. (2002b), the expected cumulative regret of EXP3 could be upper bounded by $2Q\sqrt{(e-1)LK\ln(K)}$, where $Q$ is the maximum absolute sum of rewards in any epoch, $L$ is the number of rounds and $K$ is the number of arms. Under our setting, we can set $Q = H_0, L = \lceil T/H_0\rceil$ and $K = |J| = O(\ln(H_0))$. So we could bound Quantity (II) as:

$$\mathbb{E}\left[\sum_{i=1}^{\lceil T/H_0\rceil} \sum_{t=(i-1)H_0+1}^{\min\{T,iH_0\}} f_t(v_t(H^*)) - f_t(v_t(H_{j_i}))\right] \leq 2\sqrt{e-1}H_0\sqrt{\frac{T}{H_0}|J|\ln(|J|)} = \tilde{O}(\sqrt{TH_0})$$

$$= \tilde{O}\left(T^{\frac{p_u+3}{p_u+4}}\right) = \tilde{O}\left(T^{\frac{p_u+2}{p_u+3}} T^{\frac{1}{(p_u+3)(p_u+4)}}\right) = \tilde{O}\left(T^{\frac{p_u+2}{p_u+3}} c(T)^{\frac{1}{p_u+3}}\right), \tag{17}$$

---
**Algorithm 3** Zooming TS algorithm with Double Restarts

---
**Input:** Time horizon $T$, space $A$, upper bound $p_u \leq p_c$.
**Initialization:** the (top) epoch size $H_0 = \lceil T^{(p_u+2)/(p_u+4)} \rceil$, $N = \lceil \log_2(H_0) \rceil + 1$, $J = \{H_i = \lceil H_0/2^{i-1} \rceil\}_{i=1}^N$.

1: Initialize the exponential weights $w_j(1) = 1$ for $j = 1, \ldots, |J|$.

2: Initialize the exploration parameter for the EXP3 algorithm as $\alpha = \min\left\{1, \sqrt{\frac{|J|\log(|J|)}{(e-1)\lceil T/H_0 \rceil}}\right\}$.

3: **for** $i = 1$ **to** $\lceil T/H_0 \rceil$ **do**

4:     Update probability distribution for selecting candidates in $J$ based on EXP3 as:

$$p_j(i) = \frac{\alpha}{|J|} + (1-\alpha)\frac{w_j(i)}{\sum_{k=1}^{|J|} w_k(i)}, \ j = 1, \ldots, |J|.$$

5:     Pull $j_i$ from $\{1, 2, \ldots, |J|\}$ according to the probability distribution $\{p_j(i)\}_{j=1}^{|J|}$.

6:     Run Zooming TS algorithm with Restarts using the (bottom) epoch size $H_{j_i}$ for $t = (i-1)H_0 + 1$ **to** $\min\{T, iH_0\}$, and collect the pulled arm $v_t(H_{j_i})$ and reward $Y_t$ at each iteration.

7:     Update components in EXP3: $r_j(i) = 0$ for all $j \neq j_i$; $r_j(i) = \sum_{k=(i-1)H_0+1}^{\min\{T,iH_0\}} Y_k/p_j(i)$ if $j = j_i$, and then

$$w_j(i+1) = w_j(i)\exp\left(\frac{\alpha}{|J|}r_j(i)\right), \ j = 1, \ldots, |J|.$$

8: **end for**

---

where we have the last equality since we assume that $c(T) \geq T^{1/(p_u+4)}$. Therefore, we have finished the proof for this case. (2) If $H^{\mathrm{opt}} = (T/c(T))^{(p_u+2)/(p_u+3)} > H_0$, which is equivalent to

$$\left(\frac{T}{c(T)}\right)^{\frac{p_u+2}{p_u+3}} > H_0 \Leftrightarrow \left(\frac{T}{c(T)}\right)^{\frac{p_u+2}{p_u+3}} > T^{\frac{p_u+2}{p_u+4}} \Leftrightarrow c(T) < T^{\frac{1}{p_u+4}},$$

then we know that $H^{\mathrm{opt}}$ is greater than all candidates in $J$, which means that we could not bound the Quantity (I) based on the previous argument. By simply using $H^* = H_0$, it holds that

$$\text{Quantity (I)} = \tilde{O}\left(H_0 c(T) + T \cdot H_0^{-\frac{1}{p_u+2}}\right) = \tilde{O}\left(T^{\frac{p_u+3}{p_u+4}}\right).$$

For Quantity (II), based on Eqn. equation 17, we have

$$\text{Quantity (II)} = \tilde{O}\left(T^{\frac{p_u+3}{p_u+4}}\right).$$

Combining the case (1) and (2), it holds that

$$R_L(T) \leq \begin{cases} T^{\frac{p_u+2}{p_u+3}}c(T)^{\frac{1}{p_u+3}}, & c(T) \geq T^{\frac{1}{p_u+4}}, \\ T^{\frac{p_u+3}{p_u+4}}, & c(T) < T^{\frac{1}{p_u+4}}. \end{cases}$$

And this concludes our proof. $\qquad\square$

# G  Analysis of Theorem 4.2

## G.1  Additional Lemma

**Lemma G.1** (Proposition 1 in Li et al. (2017)). *Define $V_{n+1} = \sum_{t=1}^{n} X_t X_t^T$, where $X_t$ is drawn IID from some distribution in unit ball $\mathbb{B}^d$. Furthermore, let $\Sigma := E[X_t X_t^T]$ be the second moment matrix, let $B, \delta_2 > 0$ be two positive constants. Then there exists positive, universal constants $C_1$ and $C_2$ such that $\lambda_{\min}(V_{n+1}) \geq B$*

*with probability at least $1 - \delta_2$, as long as*

$$n \geq \left( \frac{C_1 \sqrt{d} + C_2 \sqrt{\log(1/\delta_2)}}{\lambda_{\min}(\Sigma)} \right)^2 + \frac{2B}{\lambda_{\min}(\Sigma)}.$$

**Lemma G.2** (Theorem 2 in Abbasi-Yadkori et al. (2011)). *For any $\delta < 1$, under our problem setting in Section 3, it holds that for all $t > 0$,*

$$\left\| \hat{\theta}_t - \theta^* \right\|_{V_t} \leq \beta_t(\delta),$$

$$\forall x \in \mathbb{R}^d, |x^\top (\hat{\theta}_t - \theta^*)| \leq \|x\|_{v_t^{-1}} \beta_t(\delta),$$

*with probability at least $1 - \delta$, where*

$$\beta_t(\delta) = \sigma \sqrt{\log \left( \frac{(\lambda + t)^d}{\delta^2 \lambda^d} \right)} + \sqrt{\lambda} S.$$

In this subsection we denote $\alpha^*(\delta) := \beta_T(\delta)$.

**Lemma G.3** (Filippi et al. (2010)). *Let $\lambda > 0$, and $\{x_i\}_{i=1}^t$ be a sequence in $\mathbb{R}^d$ with $\|x_i\| \leq 1$, then we have*

$$\sum_{s=1}^t \|x_s\|_{V_s^{-1}}^2 \leq 2 \log \left( \frac{\det(V_{t+1})}{\det(\lambda I)} \right) \leq 2d \log \left( 1 + \frac{t}{\lambda} \right),$$

$$\sum_{s=1}^t \|x_s\|_{V_s^{-1}} \leq \sqrt{T \left( \sum_{s=1}^t \|x_s\|_{V_s^{-1}}^2 \right)} \leq \sqrt{2dt \log \left( 1 + \frac{t}{\lambda} \right)}.$$

**Lemma G.4** (Agrawal & Goyal (2013)). *For a Gaussian random variable $Z$ with mean $m$ and variance $\sigma^2$, for any $z \geq 1$,*

$$P(|Z - m| \geq z\sigma) \leq \frac{1}{\sqrt{\pi} z} e^{-z^2/2}.$$

**Lemma G.5** (Adapted from Lemma G.2). *For any $\delta < 1$, under our problem setting in Section 3 with the regularization hyper-parameter $\lambda \in [\lambda_{\min}, \lambda_{\max}] (\lambda_{\min} > 0)$, it holds that for all $t > 0$,*

$$\left\| \hat{\theta}_t - \theta^* \right\|_{V_t} \leq \beta_t(\delta),$$

$$\forall x \in \mathbb{R}^d, |x^\top (\hat{\theta}_t - \theta^*)| \leq \|x\|_{V_t^{-1}} \beta_t(\delta),$$

*with probability at least $1 - \delta$, where*

$$\beta_t(\delta) = \sigma \sqrt{\log \left( \frac{(\lambda_{\min} + t)^d}{\delta^2 \lambda_{\min}^d} \right)} + \sqrt{\lambda_{\max}} S.$$

*Proof.* The proof of this Lemma is trivial given Lemma G.2. For any $\lambda \in [\lambda_{\min}, \lambda_{\max}]$, according to Lemma G.2 it holds that, for all $t > 0$,

$$\left\| \hat{\theta}_t - \theta^* \right\|_{V_t} \leq \beta_t(\delta),$$

$$\forall x \in \mathbb{R}^d, |x^\top (\hat{\theta}_t - \theta^*)| \leq \|x\|_{V_t^{-1}} \beta_t(\delta),$$

with probability at least $1 - \delta$, where

$$\beta_t(\delta) = \sigma \sqrt{\log \left( \frac{(\lambda + t)^d}{\delta^2 \lambda^d} \right)} + \sqrt{\lambda} S \leq \sigma \sqrt{\log \left( \frac{(\lambda_{\min} + t)^d}{\delta^2 \lambda_{\min}^d} \right)} + \sqrt{\lambda_{\max}} S.$$

$\square$

## G.2 Proof of Theorem 4.2

Recall the partition of the cumulative regret as:

$$R(T) = \underbrace{\mathbb{E}\left[\sum_{t=1}^{T_1}\left(\mu(x_{t,*}^\top\theta^*) - \mu(x_t^\top\theta^*))\right)\right]}_{\text{Quantity (A)}} + \underbrace{\mathbb{E}\left[\sum_{t=T_1+1}^{T}\left(\mu(x_{t,*}^\top\theta^*) - \mu(x_t(\alpha^*(t)|\mathcal{F}_t^*)^\top\theta^*))\right)\right]}_{\text{Quantity (B)}}$$

$$+ \underbrace{\mathbb{E}\left[\sum_{t=T_1+1}^{T}\left(\mu\left(x_t(\alpha^*(t)|\mathcal{F}_t^*)^\top\theta^*\right) - \mu(x_t(\alpha^*(t)|\mathcal{F}_t)^\top\theta^*))\right)\right]}_{\text{Quantity (C)}}$$

$$+ \underbrace{\mathbb{E}\left[\sum_{t=T_1+1}^{T}\left(\mu\left(x_t(\alpha^*(t)|\mathcal{F}_t)^\top\theta^*\right) - \mu(x_t(\alpha(i_t)|\mathcal{F}_t)^\top\theta^*))\right)\right]}_{\text{Quantity (D)}}.$$

For Quantity (A), it could be easily bounded by the length of warming up period as:

$$\mathbb{E}\left[\sum_{t=1}^{T_1}\left(\mu(x_{t,*}^\top\theta^*) - \mu(x_t^\top\theta^*))\right)\right] \le T_1 = O\left(T^{\frac{2}{p+3}}\right) \le O\left(T^{\frac{p+2}{p+3}}\right). \tag{18}$$

For Quantity (B), it depicts the cumulative regret of the contextual bandit algorithm that runs with the theoretical optimal hyperparameter $\alpha^*(t)$ all the time. Therefore, if we implement any state-of-the-arm contextual generalized linear bandit algorithms (e.g. Filippi et al. (2010); Li et al. (2010; 2017)), it holds that

$$\mathbb{E}\left[\sum_{t=T_1+1}^{T}\left(\mu(x_{t,*}^\top\theta^*) - \mu(x_t(\alpha^*(t)|\mathcal{F}_t^*)^\top\theta^*))\right)\right] \le \tilde{O}(\sqrt{T-T_1}) = \tilde{O}(\sqrt{T}). \tag{19}$$

For Quantity (C), it represents the cumulative difference of regret under the theoretical optimal hyperparameter combination $\alpha^*(t)$ with two lines of history $\mathcal{F}_t$ and $\mathcal{F}_t^*$. Note for most GLB algorithms, the most significant hyperparameter is the exploration rate, which directly affect the decision-making process. Regarding the regularization hyperparameter $\lambda$, it is used to make $V_t$ invertible and hence would be set to 1 in practice. And in the long run it would not be influential. Moreover, there is commonly no theoretical optimal value for $\lambda$, and it could be set to an arbitrary constant in order to obtain the $\tilde{O}(\sqrt{T})$ bound of regret. For theoretical proof, this hyperparameter ($\lambda$) is also not significant: for example, if the search interval for $\lambda$ is $[\lambda_{\min}, \lambda_{\max}]$, then we can easily modify the Lemma G.3 as:

$$\sum_{s=1}^{t}\|x_s\|_{V_s^{-1}}^2 \le 2\log\left(\frac{\det(V_{t+1})}{\det(\lambda I)}\right) \le 2d\log\left(1 + \frac{t}{\lambda_{\min}}\right),$$

$$\sum_{s=1}^{t}\|x_s\|_{V_s^{-1}} \le \sqrt{T\left(\sum_{s=1}^{t}\|x_s\|_{V_s^{-1}}^2\right)} \le \sqrt{2dt\log\left(1 + \frac{t}{\lambda_{\min}}\right)}.$$

We will offer a more detailed explanation to this fact in the following proof of bounding Quantity (C). Furthermore, other parameters such as the stepsize in a loop of gradient descent will not be crucial either since the final result would be similar after the convergence criterion is met. Therefore, w.l.o.g we would only assume there is only one exploration rate hyperparameter here to bound Quantity (C). Recall that $\alpha(t)$ is the combination of all hyperparameters, and hence we could denote this exploration rate hyperparameter as $\alpha(t)$ in this part since there is no more other hyperparameter. Here we would use LinUCB and LinTS for the detailed proof, and note that regret bound of all other UCB and TS algorithms could be similarly deduced.

We first reload some notations: recall we denote $V_t = \lambda I + \sum_{i=1}^{t-1} x_i x_i^\top, \hat{\theta}_t = V_t^{-1} \sum_{i=1}^{t-1} x_i y_i$ where $x_t$ is the arm we pulled at round $t$ by using our tuned hyperparameter $\alpha(i_t)$ and the history based on our framework all the time. And we denote

$$X_t = \arg\max_{x \in \mathcal{X}_t} x^\top \hat{\theta}_t + \alpha^*(t) \|x\|_{V_t^{-1}}$$

Similarly, we denote $\tilde{V}_t = \lambda I + \sum_{i=1}^{t-1} \tilde{X}_i \tilde{X}_i^\top, \tilde{\theta}_t = \tilde{V}_t^{-1} \sum_{i=1}^{t-1} \tilde{X}_i \tilde{y}_i$, where $\tilde{X}_t$ is the arm we pulled by using the theoretical optimal hyperparameter $\alpha^*(t)$ under the history of always using $\{\alpha^*(s)\}_{s=1}^{t-1}$, and $\tilde{y}_t$ is the corresponding payoff we observe at round $t$. Therefore, it holds that,

$$\tilde{X}_t = \arg\max_{x \in \mathcal{X}_t} x^\top \tilde{\theta}_t + \alpha^*(t) \|x\|_{\tilde{V}_t^{-1}} .$$

By using these new definitions, the Quantity (C) could be formulated as:

$$\underbrace{\mathbb{E}\left[\sum_{t=T_1+1}^{T} \left(\mu\left(x_t(\alpha^*(t)|\mathcal{F}_t^*)^\top \theta^*\right) - \mu(x_t(\alpha^*(t)|\mathcal{F}_t)^\top \theta^*)\right)\right]}_{\text{Quantity (C)}} = \mathbb{E}\left[\sum_{t=T_1+1}^{T} \mu(\tilde{X}_t^\top \theta^*) - \mu(X_t^\top \theta^*)\right]$$

For LinUCB, since the Lemma G.2 holds for any sequence $(x_1, \ldots, x_t)$, and hence we have that with probability at least $1 - \delta$,

$$\left\|\hat{\theta} - \theta\right\|_{V_t} \le \beta_t(\delta) \le \alpha^*(T, \delta), \tag{20}$$

where

$$\beta_t(\delta) = \sigma\sqrt{\log\left(\frac{(\lambda + t)^d}{\delta^2 \lambda^d}\right)} + \sqrt{\lambda} S = \alpha^*(t).$$

And we will omit $\delta$ for simplicity. For LinUCB, we have that

$$X_t^\top \hat{\theta}_t + \alpha^*(t) \|X_t\|_{V_t^{-1}} \ge \tilde{X}_t^\top \hat{\theta}_t + \alpha^*(t) \left\|\tilde{X}_t\right\|_{V_t^{-1}}$$
$$\ge \tilde{X}_t^\top \theta^* + \alpha^*(t) \left\|\tilde{X}_t\right\|_{V_t^{-1}} + \tilde{X}_t^\top (\hat{\theta}_t - \theta^*) \ge \tilde{X}_t^\top \theta^*.$$

Therefore, it holds that

$$X_t^\top \theta^* + \alpha^*(t) \|X_t\|_{V_t^{-1}} + X_t^\top (\hat{\theta}_t - \theta^*) \ge \tilde{X}_t^\top \theta^*$$
$$X_t^\top \theta^* + 2\alpha^*(t) \|X_t\|_{V_t^{-1}} \ge \tilde{X}_t^\top \theta^*,$$

which implies that

$$(\tilde{X}_t - X_t)^\top \theta^* \le 2\alpha^*(T) \|X_t\|_{V_t^{-1}} .$$

By Lemma G.3 and choosing $T_1 = T^{2/(p+3)}$, it holds that,

$$\sum_{t=T_1+1}^{T} \|X_t\|_{V_t^{-1}} \le \sum_{t=T_1+1}^{T} \|X_t\| \sqrt{\lambda_{\min}(V_t)} = O(T \times T^{-1/(p+3)}) = O(T^{(p+2)/(p+3)}).$$

And then it holds that,

$$\sum_{t=T_1+1}^{T} \left(\tilde{X}_t^T \theta - X_t \theta\right) = \tilde{O}\left(\alpha^*(T) \sum_{t=T_1+1}^{T} \left\|\tilde{X}_t\right\|_{V_t^{-1}}\right) = \tilde{O}(T^{(p+2)/(p+3)}). \tag{21}$$

Note $\beta_t(\delta)$ contain the regularizer parameter $\lambda$, and it's often set to some constant (e.g. 1) in practice. If we tune $\lambda$ in the search interval $[\lambda_{\min}, \lambda_{\max}]$, then we can still have the identical bound as in Eqn. equation 20 by using the fact that

$$\beta_t(\delta) = \sigma\sqrt{\log\left(\frac{(\lambda + t)^d}{\delta^2 \lambda^d}\right)} + \sqrt{\lambda} S \le \sigma\sqrt{\log\left(\frac{(\lambda_{\min} + t)^d}{\delta^2 \lambda_{\min}^d}\right)} + \sqrt{\lambda_{\max}} S.$$

This result is deduced in our Lemma G.5, which implies that tuning the regularizer hyperparameter would not affect the order of final regret bound in Eqn. equation 21. Therefore, as we mentioned earlier, we could only consider the exploration rate as the unique hyperparameter for theoretical analysis.

For LinTS, we have that

$$
\begin{aligned}
X_t^\top \hat{\theta}_t + \alpha^*(T) \|X_t\|_{V_t^{-1}} Z_t &\geq \tilde{X}_t^\top \hat{\theta}_t + \alpha^*(T) \|\tilde{X}_t\|_{V_t^{-1}} \tilde{Z}_t \\
&\geq \tilde{X}_t^\top \theta^* + \alpha^*(T) \|\tilde{X}_t\|_{V_t^{-1}} \tilde{Z}_t + \tilde{X}_t^\top (\hat{\theta}_t - \theta^*) \\
&\geq \tilde{X}_t^\top \theta^* + \alpha^*(T) \|\tilde{X}_t\|_{V_t^{-1}} \tilde{Z}_t + \|\tilde{X}_t\|_{V_t^{-1}} \|\hat{\theta}_t - \theta^*\|_{V_t} \\
&\geq \tilde{X}_t^\top \theta + (\alpha^*(T)\tilde{Z}_t - \alpha^*(T)) \|\tilde{X}_t\|_{V_t^{-1}},
\end{aligned}
$$

where $Z_t$ and $Z_{t,*}$ are IID normal random variables, $\forall t$. And then we could deduce that

$$
X_t^\top \theta^* + \alpha^*(T) \|X_t\|_{V_t^{-1}} Z_t + X_t^\top (\hat{\theta}_t - \theta^*) \geq \tilde{X}_t^\top \theta + (\alpha^*(T)\tilde{Z}_t - \alpha^*(T)) \|\tilde{X}_t\|_{V_t^{-1}}
$$

$$
X_t^\top \theta^* + \alpha^*(T) \|X_t\|_{V_t^{-1}} Z_t + \alpha^*(T) \|X_t\|_{V_t^{-1}} \geq \tilde{X}_t^\top \theta + (\alpha^*(T)\tilde{Z}_t - \alpha^*(T)) \|\tilde{X}_t\|_{V_t^{-1}}
$$

$$
(\tilde{X}_t - X_t)^\top \theta^* \leq (\alpha^*(T) - \alpha^*(T)\tilde{Z}_t) \|\tilde{X}_t\|_{V_t^{-1}} + (\alpha^*(T) + \alpha^*(T)Z_t) \|X_t\|_{V_t^{-1}} := K_t
$$

where $K_t$ is normal random variable with

$$
\mathbb{E}(K_t) \leq 2\alpha(T)T^{-1/(p+3)}, \ \mathrm{SD}(K_t) \leq \sqrt{2}\alpha^* T^{-1/(p+3)}.
$$

Consequently, we have

$$
\sum_{t=T_1+1}^{T} \left( \tilde{X}_t^T \theta - X_t^T \theta \right) \leq \sum_{t=T_1+1}^{T} K_t := K
$$

$$
\mathbb{E}(K) = 2\alpha^*(T)T^{(p+2)/(p+3)} = \tilde{O}(T^{\frac{p+2}{p+3}}), \ \mathrm{SD}(K) \leq \sqrt{2}\alpha^* T^{\frac{p+1}{2p+6}} = O(T^{\frac{p+1}{2p+6}}).
$$

Based on Lemma G.4, we have

$$
P\left( \sum_{t=T_1+1}^{T} \left( \tilde{X}_t^T \theta - X_t^T \theta \right) \geq K > (2\alpha^* + \sqrt{2})T^{\frac{p+2}{p+3}} \right) \leq \frac{1}{c\sqrt{\pi}\sqrt{T}} e^{-c^2 T/2}. \tag{22}
$$

This probability upper bound is minimal and negligible, which means the bound on its expected value (Quantity (C)) could be easily deduced. Note we could use this procedure to bound the regret for other UCB and TS bandit algorithms, since most of the proof for GLB algorithms are closely related to the rate of $\sum_{t=T_1+1}^{T} \|X_t\|_{V_t^{-1}}$ and the consistency of $\hat{\theta}_t$. In conclusion, we have that Quantity (C) could be upper bounded by the order $\tilde{O}(T^{\frac{p+2}{p+3}})$.

For Quantity (D), which is the extra regret we paid for hyperparameter tuning in theory. Recall we assume $\mu(x_t(\alpha|\mathcal{F}_t)^\top \theta^*) = g_t(\alpha) + \eta_{\mathcal{F}_t,\alpha}$ for some time-dependent Lipschitz function $g_t$. And $(\eta_{\mathcal{F}_t,\alpha} - \mathbb{E}[\eta_{\mathcal{F}_t,\alpha}])$ is IID sub-Gaussian with parameter $\tau^2$ where $\mathbb{E}[\eta_{\mathcal{F}_t,\alpha}]$ depends on the history $\mathcal{F}_t$. Denote $\nu_{\mathcal{F}_t,\alpha} = \eta_{\mathcal{F}_t,\alpha} - \mathbb{E}[\eta_{\mathcal{F}_t,\alpha}]$ is the IID sub-Gaussian random variable with parameter $\tau^2$, then we have that

$$
y_t = g_t(\alpha(i_t)) + \nu_{\mathcal{F}_t,\alpha(i_t)} + E[\eta_{\mathcal{F}_t,\alpha(i_t)}] + \epsilon_t
$$

Since $\nu_{\mathcal{F}_t,\alpha(i_t)}, \epsilon_t$ is IID sub-Gaussian random variable independent with $\mathcal{F}_t$, we denote $\tilde{\epsilon}_{\mathcal{F}_t,\alpha(i_t)} = \nu_{\mathcal{F}_t,\alpha(i_t)} + \epsilon_t$ as the IID sub-Gaussian noise with parameter $\tau^2 + \sigma^2$. And then we have

$$
y_t = g_t(\alpha(i_t)) + E[\eta_{\mathcal{F}_t,\alpha(i_t)}] + \tilde{\epsilon}_{\mathcal{F}_t,\alpha(i_t)}, \quad \mathbb{E}(y_t) = g_t(\alpha(i_t)) + E[\eta_{\mathcal{F}_t,\alpha(i_t)}]
$$

$$
\mu(x_t(\alpha|\mathcal{F}_t)^\top \theta^*) = g_t(\alpha) + E[\eta_{\mathcal{F}_t,\alpha}].
$$

For Quantity (D), recall it could be formulated as:

$$\underbrace{\mathbb{E}\left[\sum_{t=T_1+1}^{T}\left(\mu\left(x_t(\alpha^*(t)|\mathcal{F}_t)^\top\theta^*\right)-\mu(x_t(\alpha(i_t)|\mathcal{F}_t)^\top\theta^*))\right)\right]}_{\text{Quantity (D)}}.$$

Since both terms in Quantity (D) are based on the same line of history $\mathcal{F}_t$ at iteration $t$, and the value of $E[\eta_{\mathcal{F}_t,\alpha}]$ only depends on the history filtration $\mathcal{F}_t$ but not the value of $\alpha$. Therefore, it holds that

$$\underbrace{\mathbb{E}\left[\sum_{t=T_1+1}^{T}\left(\mu\left(x_t(\alpha^*(t)|\mathcal{F}_t)^\top\theta^*\right)-\mu(x_t(\alpha(i_t)|\mathcal{F}_t)^\top\theta^*))\right)\right]}_{\text{Quantity (D)}}=\sum_{t=T_1+1}^{T}g_t(\alpha^*(t))-\mathbb{E}[g_t(\alpha(i_t))]$$
$$\leq\sum_{t=T_1+1}^{T}\sup_{\alpha\in A}g_t(\alpha)-\mathbb{E}[g_t(\alpha(i_t))].$$

Therefore, Quantity (D) could be regarded as the cumulative regret of a non-stationary Lipschitz bandit and the noise is IID sub-Gaussian with parameter $\tau_0^2=(\tau^2+\sigma^2)$. We assume that, under the *switching* environment, the Lipschitz function $g_t(\cdot)$ would be piecewise stationary and the number of change points is of scale $\tilde{O}(1)$. Therefore, Quantity (D) can be upper bounded the cumulative regret of our Zooming TS algorithm with restarted strategy given $c(T)=\tilde{O}(1)$. By choosing $T_2=(T-T_1)^{(p+2)/(p+3)}=\Theta(T^{(p+2)/(p+3)})$, and according to Theorem 4.1, it holds that,

$$\sum_{t=T_1+1}^{T}\sup_{\alpha\in A}g_t(\alpha)-\mathbb{E}[g_t(\alpha(i_t))]\leq\tilde{O}\left(T^{\frac{p+2}{p+3}}\right). \tag{23}$$

By combining the results deduced in Eqn. equation 18, Eqn. equation 19, Eqn. equation 21 (or Eqn. equation 22) and Eqn. equation 23, we finish the proof of Theorem 4.2 for linear bandits. For generalized linear bandits, under the default and standard assumption in the generalized linear bandit literature that the derivative of $\mu(\cdot)$ could be upper bounded by some constant given $|x|\leq S$, the regret could be bounded by further multiplying a constant in the same order.

$\square$

