# OpenReview forum: "Online Continuous Hyperparameter Optimization for Generalized Linear Contextual Bandits"
_TMLR — Accepted by TMLR_

### Review · Reviewer_ZP1P · 2024-02-09

**Summary Of Contributions:**

## Main Summary

Most online learning algorithms (whether that be UCB, LinUCB, etc) rely on hyperparameters implicitly defined in the algorithm statement and are required to satisfy precise rates to obtain the optimal theoretical guarantees.  However, in practice, the empirical performance of these algorithms explicitly depends on the chosen hyperparameters, and often it is the case that the theoretically prescribed ones lead to worse empirical performance.  In a nutshell, these hyperparameters should be chosen in an instance-dependent (or even possibly time-dependent) way in order to achieve the best performance from a given bandit algorithm.

At a high level, the authors address this challenge of choosing the hyperparameters by introducing an online hyperparameter optimization framework, which the authors call Continuous Dynamic Tuning (CDT), applied to generalized linear contextual bandits.  The CDT framework is developed in a double-bandit scenario, also referred to as ``bandit over bandits'' where there is a parameter set that dictates the resulting arm selection given by the low-level bandit algorithm.  Their top-level algorithm leverages a novel Zooming Thompson Sampling (TS) algorithm, which dynamically adjusts to changes in the environment by restarting and zooming into promising regions of the hyperparameter space, thereby handling non-stationary Lipschitz bandit problems efficiently. The framework's scalability is demonstrated through theoretical guarantees that depend on a time-dependent zooming dimension.

Lastly, the authors complement the theoretical results with experimental results.  These experimental results validate the framework's effectiveness and superiority over traditional tuning methods across synthetic datasets with applications in generalized linear bandit problems. The results showcase CDT's ability to outperform other hyperparameter tuning methods and achieve sublinear regret, highlighting its practicality and robustness in various settings.

## Model

The main model is defined over-generalized linear contextual bandits.  Over rounds $t$ the algorithm is faced with a set $A_t$ of potential actions each associated with a $d$ dimensional feature vector.  After selecting an action $a_t$ they observe a reward $y_t = \mu(x_t^\top \theta^\star) + \epsilon_t$ where $\epsilon_t$ is independent noise, $x_t$ is the feature vector for the selected action $a_t$, $\theta^\star$ unknown, and $\mu$ is the so-called link function.  The common regret-optimal algorithm in this setting, LinUCB, selects actions via estimating the parameter $\theta^\star$ via $\hat{\theta}$ using say regression, and adding on an exploration bonus proportional to $\alpha(t)$, a hyperparameter.  In order to address the choice of the hyperparameter $\alpha(t)$ the authors introduce another level, the so-called top-level, where the algorithm can pick $\alpha(t)$ in some parameter set of dimension $p$.  The observed reward then is taken to be:

$$ \mu(x_t(\alpha)^\top \theta^*) = g_t(\alpha) + \eta$$ where $\eta$ is potentially biased noise and $g_t(\alpha)$ just denotes the dependence on $\alpha$ since $\alpha$ dictates the hyperparameter.  The performance of the algorithms is measured in terms of regret, cumulative loss over the $T$ rounds to the best fixed parameter $\alpha$ / best fixed action $a$ in hindsight.

## Main Contributions

The main theoretical contributions in the paper are as follows:

- (Zooming TS Algorithm with Restarts) In the first main theoretical contribution the authors provide a novel modification of the well-studied zooming algorithm for Lipschitz bandits which includes $(i)$ Thompson Sampling in the action selection, and $(ii)$ Restarts to account for the dynamic environment.  In Theorem 4.1 the authors provide a regret guarantee for this approach scaling with respect to the so-called zooming dimension in the model (of the order $T^{(p+2)/(p+3)}$) where $p$ is the zooming dimension.
- (Continuous Hyperparameter Optimization) In the second contribution the authors use the previous algorithm as a subroutine for hyperparameter optimization of contextual bandit algorithms.  In particular, by decomposing the regret into four different quantities and dealing with the counterfactual outcomes had a fixed hyperparameter being chosen across the entire time horizon, in Theorem 4.2 the authors provide a regret guarantee of this approach on the order of $T^{(p+2)/(p+3)}$ where $p$ is the covering dimension.

**Audience:**

Yes

**Broader Impact Concerns:**

The work provides a general-purpose algorithm for online continuous hyperparameter tuning.  The authors note that $(i)$ the work provides a general purpose algorithm outside of any particular application and $(ii)$ researchers utilizing the method should adhere to ethical standards for their specific context.

**Claims And Evidence:**

Yes

**Requested Changes:**

## Requested Changes

See the main weaknesses highlighted above.  With changes to the writing style to help the presentation to be much clearer, as well as better defining the model of hyperparameter optimization the theoretical results could be easier to appreciate.

## Minor Comments
The paper needs some proofreading to clear up some of the issues with the writing, but here are some issues that I noticed:
- The entire second paragraph on page 2 is unclear
- ``Without requiring a pre-defined set of hyperparameter candidate'' on page 2 seems like a misformed comment, especially since your algorithm similarly requires a feasible parameter set
- ``To the best of our knowledge'' is a bit redundant on page 2
- ``However, ..., both works have several limitations that could be improved'' on page 3
- The point re. the regret bound on order $O(\sqrt{MT} + M R_{max})$ on page 3 was interesting and could be expanded upon later in the text to help interpret the final bounds
- Typo on top of page 4 should be subset instead of inclusion in $R^d$
- Frequently refer to the optimal values $\alpha^\star(t)$ although that is never formally defined
- Did not discuss the main differences between Algorithm 1 and the standard zooming algorithm (all of the differences are in line 10 with an index $I_t(v)$ which is not defined in the pseudocode)
- ``reload some notations'' on page 6
- ``fix distribution'' on page 8
- ``work:'' on page 8


## Questions
- Do you have any intuition for the lower bounds in Theorems 4.1 and 4.2?  Clearly Theorem 4.1 is optimal up to logarithmic factors for the case when the environment is not dynamic.  However, it is unclear whether the dynamic changes should make the dependence on the zooming dimension to be worse.  Similarly, I am surprised that in Theorem 4.2 we are taking a problem whose regret guarantee is initially $\sqrt{T}$ and changing that to one which is exponential in $p$?  At this point, for many hyperparameters, why not just use the theoretical ones (which might be too large) but still retain the reasonable regret guarantees?
- An obvious philosophical question from reading Section 4.2 is whether it makes sense to propose an algorithm (which has hyperparameters) to optimize a sub-algorithm's hyperparameters?

**Strengths And Weaknesses:**

## Strengths
The strengths of the paper are highlighted through its theoretical contributions, specifically in two main areas:
- Zooming Thompson Sampling with Restarts: The authors introduce a novel algorithm coined Zooming Thompson Sampling with restarts to address the dynamic Lipschitz bandit environment.  The method combines the empirical strengths of Thompson Sampling, restarts, and zooming, to allow the algorithm to achieve sublinear regret guarantees.
- Theoretical Guarantees for Hyperparameter Optimization: The paper provides strong theoretical foundations for online continuous hyperparameter optimization, a significant advancement in the field of contextual bandits. This framework formulates the online hyperparameter tuning problem as a non-stationary Lipschitz continuum-armed bandit problem, integrating the previous Zooming Thompson Sampling algorithm with restarts to adaptively refine the hyperparameter space and focus on promising regions for reward improvement.

## Weaknesses
- Writing Clarity: The writing of the paper is incredibly difficult to follow.
    - First off, the introduction needs to better highlight the main contributions in the paper (either through bullets or a more streamlined presentation).  The second and third paragraphs are a bit confusing, especially in hindsight as they seem to point out issues in the related literature that the current paper also doesn't account for (well-defined parameter sets, not recovering the minimax optimal regret guarantees for the sub-problem, etc).
    - Later in the definition of the model the interdependence between the selection of the hyperparameter $\alpha_t$ and the ``optimal choice'' $\alpha^*$ is never fully described.  This seems important for the point of the paper, since it is unclear that a fixed hyperparameter $\alpha^*$ truly is optimal.  This makes it hard to fully appreciate the regret guarantees highlighted in Theorem 4.2. Similarly, Section 4.1 introduces a completely new model / notation, that doesn't match the earlier notation established.
    - The relationship between the proposed Algorithm 1 vs the original Zooming Algorithm is not explicitly stated, and only defined implicitly with a small change in line 10 in the pseudocode for the action selection.
- Theoretical Guarantees: The theoretical guarantees of the paper seem quite weak.
    - Theorem 4.1: The regret guarantee in the stationary setting is already known to be $T^{(p+1)/(p+2)}$ where $p$ is chosen to be the zooming dimension of the problem.  The guarantee for the dynamic setting is done through a fixed restart schedule and the regret is essentially just an application of the pidgeonhole principle, requiring the restarts to be tuned based on the number of times the environment changes.  This feels like a unreasonable assumption in practice (especially since the ``need'' for the dynamic environment or considering a fixed number of changepoints in the hyperparameter optimization setting is not well motivated or understood).
    - Theorem 4.2: The authors state in  ``The substantial difficulty of our problem...'' comments that they recover the $\sqrt{T}$ regret guarantee in the case of generalized linear bandits with the addition of hyperparameter tuning.  However, to my understanding Theorem 4.2 will still propose a rate of $O(T^{(p+1) / (p+2)})$ where $p$ is the dimension of the hyperparameters.  You could consider padding the hyperparameters with additional dummy ones that have no impact on the underlying problem and the proposed guarantees will scale to $O(T)$, even though just using the theoretically provided ones will have sublinear regret.

---

### Review · Reviewer_THCC · 2024-02-19

**Summary Of Contributions:**

This paper investigates the hyperparameter optimization problem for generalized linear bandits (GLB), which is well-motivated since the application of GLB significantly depends on at least one hyperparameter, i.e., the exploration parameter. To this end, the authors formulate the hyperparameter optimization problem as a non-stationary (but piecewise-stationary) continuum-armed Lipschitz bandit problem, and propose a Zooming TS algorithm with Restarts for this problem, i.e., Algorithm 1 in this paper. Then, the authors propose an efficient bandit-over-bandit (BOB) framework, in which the top layer first utilizes Algorithm 1 to select the hyperparameter on the fly and runs the contextual bandit algorithm with the selected hyperparameter. As proved by the authors, for UCB and TS GLB algorithms with exploration hyperparameters, the proposed algorithm can achieve an expected regret bound of $\\tilde{O}(T^{(p+2)/(p+3)})$, where $p$ is the number of hyperparameters. Moreover, experimental results on both synthetic and real datasets are presented to demonstrate the performance of the proposed algorithm.

**Audience:**

Yes

**Broader Impact Concerns:**

No concerns.

**Claims And Evidence:**

Yes

**Requested Changes:**

There also exist some minor comments.
1) Page 1: a related work [1] about logistic bandit algorithms with binary rewards is missing.
2) Pages 1 and 23: IID or iid should be consistent.
3) Pages 5 and 6: w.l.o.g or w.l.o.g. should be consistent.
4) Theorem 4.1: there exists a redundant space before “when ...”.
5) Table 1: the authors should briefly explain why the proposed algorithm is the slowest one.

[1] Zhang et al. Online Stochastic Linear Optimization under One-bit Feedback. In ICML, 2016.

**Strengths And Weaknesses:**

#Strengths
1) The proposed algorithm does not need a pre-defined set of candidates as the existing methods in Bouneffouf & Claeys (2020) and Ding et al. (2022b).
2) Experimental results on both synthetic and real datasets are presented to demonstrate the performance of the proposed algorithm.

#Weaknesses
1) The Zooming TS algorithm with Restarts seems to be the most critical part of the Continuous Dynamic Tuning (CDT) framework. Although this extension of the classical Zooming is new, it seems to be a straightforward combination with the restart technique, the novelty of which is limited.
2) In general, the proposed algorithm only achieves an expected regret bound of $\\tilde{O}(T^{(p+2)/(p+3)})$, where $p$ is the number of hyperparameters. Even if $p=1$, this bound reduces to $\\tilde{O}(T^{3/4})$, which is worse than the $\\tilde{O}(\\sqrt{T})$ regret bound achieved by ideal parameters.

---

### Review · Reviewer_hXyJ · 2024-02-23

**Summary Of Contributions:**

This paper studies the online hyperparameter optimization for contextual bandits. The authors propose a bandit-over-bandit framework named Continuous Dynamic Tuning (CDT) to reduce the problem to a non-stationary continuum-armed bandit problem. The authors provide theoretical guarantees for the proposed algorithm; the authors also run experiments to show the efficacy of the proposed algorithm.

**Audience:**

Yes

**Claims And Evidence:**

Yes

**Requested Changes:**

I recommend the authors add discussion/material regarding the two weakness points I mentioned above.

**Strengths And Weaknesses:**

Strengths: This paper is generally well-written and easy to follow. The paper studies a practically important question of online hyperparameter selection in contextual bandits. The authors have proposed a new algorithm and proves its regret upper bound.

Weaknesses: I'm mainly concerned with the following two points.
1. The authors emphasize that their algorithm is the first hyperparameter tuning framework with a continuous set of hyperparameters. However, it seems that one can use a covering argument to discretize a continuous set of parameters and then apply a CORRAL-type of algorithm to conduct hyperparameter selection. One can adjust the radius of the covering argument based on the time horizon T to prove the regret bound.
2. The lower bound analysis is missing. Thus, it's not clear how good is the proposed algorithm.

---

### Decision · Action_Editor_adf6 · 2024-03-30

**Recommendation:** Accept as is

**Comment:**

This paper studies the hyperparameter optimization problem for generalized linear bandits, by introducing an online hyperparameter optimization framework named Continuous Dynamic Tuning. The authors propose Zooming Thompson sampling algorithm as the top-level algorithm which adjusts to the changes in the environment by restarting.

The reviewers agree that this paper studies an practical question (hXyJ), overcomes some limitations of prior works (THCC), and has provable regret guarantees (ZP1P), backed by experimental results (THCC).

During the discussion stage most concerns of the reviewers have been addressed. One remaining concern is Reviewer ZP1P's confusion about the mathematical definition of theoretically optimal \alpha^* (e.g. page 32). Please define it formaly in the final version.

**Audience:**

Yes

**Claims And Evidence:**

Yes